METHODS

# Analytical and computational solution for the estimation of SNP-heritability in biobank-scale and distributed datasets

**Guo-An Qi[1,2☯], Qi-Xin Zhang[3☯], Jingyu Kang[4], Tianyuan Li[4], Xiyun Xu[4], Zhe Zhang[5], Zhe Fan[6], Siyang Liu[6], Guo-Bo Chen[2,7]***

**1** Institute of Bioinformatics, Zhejiang University, Hangzhou, Zhejiang, China, **2** Center for Laboratory Medicine, Department of Genetic and Genomic Medicine, and Clinical Research Institute, Zhejiang Provincial People's Hospital, People's Hospital of Hangzhou Medical College, Hangzhou, Zhejiang, China, **3** Department of Epidemiology, School of Public Health, Zhejiang Chinese Medical University, Hangzhou, Zhejiang, China, **4** School of Mathematics and Statistics and Research Institute of Mathematical Sciences (RIMS), Jiangsu Provincial Key Laboratory of Educational Big Data Science and Engineering, Jiangsu Normal University, Xuzhou, Jiangsu, China, **5** Department of Animal Science, College of Animal Sciences, Zhejiang University, Hangzhou, Zhejiang, China, **6** School of Public Health (Shenzhen), Shenzhen Campus of Sun Yat-sen University, Shenzhen, Guangdong, China, **7** Key Laboratory of Endocrine Gland Diseases of Zhejiang Province, Hangzhou, Zhejiang, China

☯ These authors contributed equally to this work.
* chenguobo@gmail.com

## Author summary

For a complex trait, heritability ($h^2$) gives the genetic determination of its variation. Given the emergence of biobank-scale data, a more powerful method is needed to estimate $h^2$. Based on the framework of Haseman-Elston regression (RHE-reg), we integrate a fast randomization algorithm to estimate $h^2$, and RHE-reg can tackle biobank-scale data, such as UK Biobank (UKB), very efficiently. Furthermore, we present an analytical solution that balances computational cost and precision of the estimation, a property that is important in dealing with biobank-scale data. We investigated the performance of the RHE-reg in simulated data and also applied it for 81 UKB quantitative traits; as tested in UKB data of nearly 300,000 unrelated individuals, it took on average about 4.5 hours to complete an estimation when used 10 CPUs. We extended the application of RHE-reg into distributed datasets when privacy is not compromised. As shown in UKB and simulated data the performance of RHE-reg was accurate in estimating $h^2$. The software for estimating SNP-heritability for biobank-scale data is released.

## Abstract

Estimation of heritability has been a routine in statistical genetics, in particular with the increasing sample size such as biobank-scale data and distributed datasets, the latter of which has increasing concerns of privacy. Recently a randomized Haseman-Elston regression (RHE-reg) has been proposed to estimate SNP-heritability, and given sufficient iteration ($B$) RHE-reg can tackle biobank-scale data, such as UK

**Data availability statement:** All codes for simulation study and practical protocol are available on GitHub (https://github.com/gc5k/gear2). The genotype-phenotype data used in our analyses are available from UK Biobank (https://www.ukbiobank.ac.uk). The UKB data can be accessed following successful application at https://www.ukbiobank.ac.uk/enable-your-research/apply-for-access.

**Funding:** This work was supported by the National Natural Science Foundation of China (32272832 to ZZ, and 31771392 to GBC), Shenzhen Basic Research Foundation (20220818100717002 to SL), Guangdong Basic and Applied Basic Research Foundation (2022B1515120080 to SL). The funders had no role in study design, data collection and analysis, decision to publish, or preparation of the manuscript.

**Competing interests:** The authors have declared that no competing interests exist.

Biobank (UKB), very efficiently. In this study, we present an analytical solution that balances iteration $B$ and RHE-reg estimation, which resolves the convergence of the proposed RHE-reg in high precision. We applied the method for 81 UKB quantitative traits and estimated their SNP-heritability and test statistics precisely. Furthermore, we extended RHE-reg into distributed datasets and demonstrated their utility in real data application and simulated data.

## Introduction

Estimating heritability has been one of the central tasks in statistical genetics [1]. Given the increasing sequencing capability, high-throughput genetic data have been emerging in the form of biobank-scale [2], which challenges statistical computation, in particular, such as the estimation of heritability for complex traits. Conventional methods, such as REML, for estimating heritability, like the linear mixed model, often takes computational cost of $\mathcal{O}(n^2m + n^3)$, where $n$ is the sample size and $m$ is the number of markers. These costs can become infeasible in the context of biobank-scale data. Haseman-Elston regression (HE-reg) was originally proposed for linkage analysis [3]. After the nuclear correlation between sib pairs is replaced by the linkage disequilibrium (LD) for unrelated samples, the modified HE-reg can be used to estimate heritability and is much faster than REML [4]. Given that $m$ is often greater than $n$ given the current data, any calculation that is upon genetic relationship matrix (GRM) will be unfavorable even for HE-reg. To reduce the computational cost of estimating heritability, a randomized estimation of heritability has been introduced [5], called randomized Haseman-Elston regression (RHE-reg), which is a promising method that can be used for both single-trait and bi-trait analyses [6,7].

RHE-reg is built on a hybrid framework, which has favored analytical properties of the Haseman-Elston regression and the feasible computational cost of $\mathcal{O}(nmB)$ for biobank-scale data; $B$ is the round of iteration for RHE-reg. As pointed out by a recent systematic review, iteration control poses one of the challenges for RHE-reg [8]. However, the original report by Wu and Sankararaman did not give a clear solution for the round of iteration [5]. In this study, we investigated RHE-reg and found an analytical procedure to control $B$, which can provide customized iteration for a given data.

Now, one of the trends is that genomic cohorts are mushroomed such as emerging non-invasive prenatal testing cohorts [9,10], but the bottleneck is how to share genomic data without compromising personal privacy [11]. As recently practiced, when genotypes have been masked in randomization, the randomized method has proven to be reliable in addressing genetic problems for distributed data, such as searching relatives [12,13]. Following this idea, it is found that after the randomization step, RHE-reg can be modified to estimate heritability for distributed datasets, reminiscent of vertical or horizontal federated learning [14].

## Method description

Wu and Sankararaman proposed a randomized implementation for the Haseman-Elston regression (RHE-reg), which dramatically reduced the computational time from

$\mathcal{O}(n^2m)$ to $\mathcal{O}(nmB)$ in dealing with $\text{tr}(\boldsymbol{K}^2)$ [5]; $\boldsymbol{K}$ is the genetic relationship matrix for $n$ individuals on $m$ markers, and see its detailed definition in the section below. It is clear that a large $B$, indicating more iteration of the presented algorithm, is helpful in improving precision, but it is unsolved how to get an estimate for $B$ and its role in determining the boundaries of key statistics, upon the standard errors of the randomized estimator [8]. This work is in general consistent with Wu and Sankararaman's work, but we present the analytical sampling variance of the estimated $h^2$ and its corresponding test statistics after correction of some technical errors in their original work. We can consequently evaluate how $B$ influences the estimation of heritability and its corresponding $z$ score, and, as data can be very large, the control of $B$ is of theoretical as well as practical importance. An analytical resolution crystallizes a computational procedure, and we further extend the method to another two new scenarios, called vertical-RHE-reg, which is a global implementation for LD score regression [15], and horizontal-RHE-reg, which enables Federated Learning but we estimate heritability in distributed data without compromise of privacy [14].

## Materials and methods

### A framework for Randomized Haseman-Elston regression (RHE-reg)

In essence, Haseman-Elston regression is a kind of method of moments (MoM) estimation for heritability, and can provide equivalent estimates of heritability for complex traits after IBD is replaced with IBS [4,11]. As we extend the work of Wu and Sankararaman [5], we similarly assume that

$$\boldsymbol{y} = \boldsymbol{X}\beta + \boldsymbol{e};\ \beta \sim N\left(0, \frac{h^2}{m}\boldsymbol{I}_m\right);\ \boldsymbol{e} \sim N\left(0, \sigma_e^2\boldsymbol{I}_n\right)$$

in which $\boldsymbol{y}$ is the standardized phenotype of the traits of interest, $\boldsymbol{X}$ is the standardized genotype matrix of $n$ individuals, $m$ is the number of double allelic markers, $\beta$ is the cumulative effect related to each of the markers, $\boldsymbol{e}$ is the residual effect, $\boldsymbol{I}_m$ is an $m \times m$ identity matrix, $h^2$ is the SNP heritability, and $\boldsymbol{I}_n$ is an $n \times n$ identity matrix, $\sigma_e^2$ is the residual variance. Under the general assumption for a polygenic trait, it is easy to see that

$$var(\boldsymbol{y}) = E(\boldsymbol{y}\boldsymbol{y}^T) - E(\boldsymbol{y})E(\boldsymbol{y}^T) = \frac{h^2}{m}\boldsymbol{X}\boldsymbol{X}^T + \sigma_e^2\boldsymbol{I}_n = h^2\boldsymbol{K} + \sigma_e^2\boldsymbol{I}_n$$

$\boldsymbol{K} = \frac{1}{m}\boldsymbol{X}\boldsymbol{X}^T$ is the genetic relationship matrix (GRM); the moment estimator, or randomized Haseman-Elston regression, is to minimize $\mathcal{Q} = \text{tr}\left\{\left[\boldsymbol{y}\boldsymbol{y}^T - \left(h^2\boldsymbol{K} + \sigma_e^2\boldsymbol{I}_n\right)\right]^2\right\}$. Of $\mathcal{Q}$, by differentiating $h^2$ and $\sigma_e^2$, respectively, we have the following normal equations:

$$\begin{bmatrix} \text{tr}(\boldsymbol{K}^2) & \text{tr}(\boldsymbol{K}) \\ \text{tr}(\boldsymbol{K}) & n \end{bmatrix} \begin{bmatrix} \widehat{h^2} \\ \widehat{\sigma_e^2} \end{bmatrix} = \begin{bmatrix} \boldsymbol{y}^T\boldsymbol{K}\boldsymbol{y} \\ \boldsymbol{y}^T\boldsymbol{I}_n\boldsymbol{y} \end{bmatrix} \tag{1}$$

The preliminary estimators for $\hat{h}^2$ and $\hat{\sigma}_e^2$ are given as

$$\begin{bmatrix} \widehat{h^2} \\ \widehat{\sigma_e^2} \end{bmatrix} = \begin{cases} \dfrac{\boldsymbol{y}^T[n\boldsymbol{K} - \text{tr}(\boldsymbol{K})\boldsymbol{I}_n]\boldsymbol{y}}{n[\text{tr}(\boldsymbol{K}^2) - n]} \\ \dfrac{\boldsymbol{y}^T[\text{tr}(\boldsymbol{K}^2)\boldsymbol{I}_n - \boldsymbol{K}\text{tr}(\boldsymbol{K})]\boldsymbol{y}}{n\text{tr}(\boldsymbol{K}^2) - n^2} \end{cases} \tag{2}$$

For ease of discussion, we now only focus on the expression without adjustment of covariates. The denominator involves $\text{tr}(\boldsymbol{K}^2)$, a high-order function for GRM. Alternatively, according to the trace property of a matrix, it can be calculated that $\text{tr}(\boldsymbol{K}^2) = \Sigma_{i,j}^n K_{i,j}^2$, a summation of the square of each element in $\boldsymbol{K}$. We proved that the expectation of

$\text{tr}\left(\boldsymbol{K}^2\right) = \frac{n(n+1)}{m_e} + n$, where $m_e$ is the effective number of markers that depicts the average squared Pearson's correlations among all genomic markers as often used for measuring linkage disequilibrium [12,16,17]; a brief sketch of how $\text{tr}\left(\boldsymbol{K}^2\right)$ can be transferred into $m_e$ is also presented in the section "Estimation for the effective number of markers" below. Therefore, the expectation for the preliminary estimator of $h^2$ is $E\left(\hat{h}^2\right) = \frac{m_e}{n^2}\left(\boldsymbol{y}^T\boldsymbol{K}\boldsymbol{y} - n\right) = \frac{\bar{r}^2_{mq}}{\bar{r}^2_m}h^2$ for a typical polygenic trait as established [4,18]; $\bar{r}^2_{mq}$ is the averaged LD between a marker and a causal variant, and $\bar{r}^2_m = \frac{1}{m_e} = \frac{\sum^m_{k,l}\rho^2_{kl}}{m^2}$ is the averaged LD between any pair of markers – including the LD of a marker with itself. At first glance at Eq 2, it seems inevitable to compute $\boldsymbol{K}$, the computational cost of which is $\mathcal{O}(n^2 m)$, a substantial cost given a large sample size, such as for UKB of about 500,000 samples. We obtain the estimate of $\text{tr}(\boldsymbol{K}^c)$ according to the properties of matrix algebra, and $c$ is the exponential index and $c$ takes the value of 1, 2, 3, or 4 upon the application in this study.

$$\begin{cases} L_{c,B} = \frac{1}{B}\sum^B_b \boldsymbol{z}^T_b\boldsymbol{K}^c\boldsymbol{z}_b \\ E\left(L_{c,B}\right) = \text{tr}(\boldsymbol{K}^c) \\ var\left(L_{c,B}\right) = \frac{2\text{tr}(\boldsymbol{K}^{2c})}{B} \end{cases} \tag{3}$$

Where $L_{c,B}$ is a linear estimator for $\text{tr}(\boldsymbol{K}^c)$, $\boldsymbol{z}_b$ is a vector of length $n$ and each element of $\boldsymbol{z}_b$ is sampled from the standard normal distribution, and $B$ is the round of iterations. Of note, the sampling variance of $L_{2,B} = \frac{2\text{tr}(\boldsymbol{K}^4)}{B}$. As will be shown below, $\text{tr}(\boldsymbol{K}^4)$ will be a plugin parameter in the analysis below, and we suggest a robust estimation of $\text{tr}\left(\boldsymbol{K}^4\right)$ from $L_{4,B} = \frac{1}{B}\sum^B_b \boldsymbol{z}^T_b\boldsymbol{K}^4\boldsymbol{z}_b$ rather than $\frac{B}{2}var\left(L_{2,B}\right)$. Eq 3 is the most innovative part in the work of Wu and Sankararaman, and it is known as Girard-Hutchinson estimation for stochastic trace estimation [19, 20]. Of note, $var\left(L_{2,B}\right) = \frac{2\text{tr}(\boldsymbol{K}^4)}{B}$, which was incorrectly derived as $var\left(L_{2,B}\right) = \frac{2\text{tr}(\boldsymbol{K}^2)}{B}$ in Wu and Sankararaman's work [5], and to fix their problem directly led to the present work.

## Randomized estimation for $h^2$ via RHE-Reg

When there is random mating, $E[\text{tr}\left(\boldsymbol{K}\right)] = n$, and substituting the expressions given as Eq 3 into Eq 2, a randomized estimator of heritability is

$$\widehat{h}^2 = \frac{\boldsymbol{y}^T\left[n\boldsymbol{K} - \text{tr}\left(\boldsymbol{K}\right)\boldsymbol{I}_n\right]\boldsymbol{y}}{n\left[L_{2,B} - n\right]} \approx \frac{\boldsymbol{y}^T\left[\boldsymbol{K} - \boldsymbol{I}_n\right]\boldsymbol{y}}{L_{2,B} - n} = \frac{\boldsymbol{y}^T\boldsymbol{K}\boldsymbol{y} - n}{L_{2,B} - n} \tag{4}$$

The component $L_{2,B} = \frac{1}{B}\sum^B_b \boldsymbol{z}^T_b\boldsymbol{K}^c\boldsymbol{z}_b$ in the denominator is no other than a shuffling nature of the estimation with $B$ rounds of resampling.

## Sampling variance of RHE-reg

Of Eq 4, we have $a = \boldsymbol{y}^T\left[\boldsymbol{K} - \boldsymbol{I}_n\right]\boldsymbol{y}$ and $b = L_{2,B} - n$, and their respective mean and variance are

$$\begin{cases} a\begin{cases} \mu_a = E\left(\boldsymbol{y}^T\boldsymbol{K}\boldsymbol{y} - n\right) = \left[\text{tr}\left(\boldsymbol{K}^2\right) - n\right]h^2 \\ \sigma^2_a = var\left(\boldsymbol{y}^T\left(\boldsymbol{K} - \boldsymbol{I}\right)\boldsymbol{y}\right) = 2\text{tr}[\boldsymbol{\Sigma}(\boldsymbol{K} - \boldsymbol{I})\boldsymbol{\Sigma}(\boldsymbol{K} - \boldsymbol{I})] \end{cases} \\ b\begin{cases} \mu_b = L_{2,B} - n = \text{tr}\left(\boldsymbol{K}^2\right) - n = \frac{n(n+1)}{m_e} \\ \sigma^2_b = \frac{2}{B}\text{tr}(\boldsymbol{K}^4) \end{cases} \end{cases}$$

The randomized estimator of $h^2$ can be seen as a ratio of $\frac{a}{b}$, in which both $a$ and $b$ are variables, and, according to Delta method, its sampling variance can be expressed as $var\left(\frac{a}{b}\right) = \frac{1}{\mu^2_b}\sigma^2_a - 2\frac{\mu_a}{\mu^3_b}cov(a, b) + \frac{\mu^2_a}{\mu^4_b}\sigma^2_b$, in which the covariance term can be zeroed out in this scenario [21]. So, we can obtain

$$var(\widehat{h^2}) = 2 \left(\frac{m_e}{n^2}\right)^2 \left(\Lambda_1 + \frac{\text{tr}(\mathbf{K}^4)}{B} \cdot h^4\right)$$

(5)

For the definition of $\Lambda_1$ please refer to the section "Estimation for key parameters". As $L_{2,B}$ is a random variable, using Taylor approximation $\hat{h}^2$ can be obtained by $E\left(\hat{h}^2\right) = E(a)E\left(\frac{1}{b}\right)$.

$E\left(\frac{1}{b}\right) \approx \frac{1}{L_{2,B}-n} - \frac{1}{(L_{2,B}-n)^2} E\left[\frac{1}{b} - (L_{2,B}-n)\right] + \frac{1}{(L_{2,B}-n)^3}\left[\frac{1}{b} - (L_{2,B}-n)\right]^2 = \frac{1}{L_{2,B}-n} + \frac{1}{(L_{2,B}-n)^3}\sigma_b^2.$

$$E(\widehat{h^2}) = E(a)E\left(\frac{1}{b}\right) = h^2 + 2\left(\frac{m_e}{n^2}\right)^2 \cdot \frac{\text{tr}(\mathbf{K}^4)}{B} \cdot h^2$$

(6)

in which the second term is the bias of the RHE-reg estimator. At the same time, we can also find the mean squared error (MSE), the summation of the sampling variance and squared bias, for $\hat{h}^2$ as below

$$MSE\left(\widehat{h^2}\right) = var(\widehat{h^2}) + \left[E(\widehat{h^2}) - h^2\right]^2 = 2\left(\frac{m_e}{n^2}\right)^2 \left(\Lambda_1 + \frac{\text{tr}(\mathbf{K}^4)}{B} \cdot h^4\right) + 4\left(\frac{m_e}{n^2}\right)^4 \cdot \left[\frac{\text{tr}\left(\mathbf{K}^4\right)}{B}\right]^2 \cdot h^4$$

(7)

In this polynomial expression, as will be shown in the simulation and real data analysis, $MSE\left(\hat{h}^2\right)$ is largely upon the sampling variance, which can be further reduced with sufficient iterations ($B$). As will be shown for UK Biobank examples, $B$ dynamically ranges from 10 to 200, even greater upon many factors.

**Constructing test statistics**

Given the estimation of heritability, we can construct the z-score statistic below:

$$z_1 = \frac{\hat{h}^2}{\hat{\sigma}_{h^2}} = \left(\frac{n^2}{\sqrt{2}m_e}\right) \frac{\hat{h}^2}{\sqrt{\Lambda_1}\sqrt{1 + \frac{\eta}{B}}}$$

(8)

in which $\eta = \frac{\text{tr}(\mathbf{K}^4)h^4}{\Lambda_1}$, a quantity that will be zeroed out after sufficient iterations, and $\hat{\sigma}_{h^2}$ can be estimated from Eq 5. Obviously, when $B$ is large enough, the optimal z score is the following:

$$z_2 = \left(\frac{n^2}{\sqrt{2}m_e}\right) \cdot \frac{1}{\sqrt{\Lambda_1}} \cdot \hat{h}^2 \ (B \to \infty)$$

(9)

There is an obvious relationship between two z scores in Eq 8 and Eq 9 (practically $B \approx 50$). Given $z_1$ we can predict optimal test statistic $z_3$ as below:

$$z_3 = z_1\sqrt{1 + \frac{\eta}{B}}$$

(10)

It means that after $B$ iteration the expectation of the test statistic is predictable in certain degree.

In summary, given $B$ iterations the test statistic observed is $z_1$, which is subject to the realized values of $m_e$, $\Lambda_1$, and $\hat{h}^2$. $z_2$ is the expected optimized test statistic when $B$ is very large and zeroed out all uncertainty due to iteration. $z_3$ is a reconstruction of $z_1$, and $\frac{z_3}{z_1} = \sqrt{1 + \frac{\eta}{B}}$, indicating how a larger $B$ seem to bring out advantage in such as a more significant p-value.

## Estimation for key parameters

There are several key quantities/parameters involved in the above equations for RHE-reg, and we present how to estimate them. These parameters are $m_e$ – effective number of markers, $tr(K^4)$ the trace of fourth-order GRM, and $\Lambda_1$.

### Estimation for the effective number of markers ($m_e$)

$$E\left[\text{tr}\left(K^2\right)\right] = E\left\{\frac{1}{m^2}\sum_{i,j}^{n}\left[\sum_{k}^{m}(x_{ik}x_{jk})\right]^2\right\} = E\left\{\frac{1}{m^2}\sum_{i,j}^{n}\left\{\left[\sum_{k}^{m}(x_{ik}x_{jk})\right]\left[\sum_{l}^{m}(x_{il}x_{jl})\right]\right\}\right\}$$

$$E\left[\text{tr}\left(K^2\right)\right] = \frac{1}{m^2}\sum_{i,j}^{n}\left\{\left[\sum_{k}^{m}(x_{ik}x_{jk})\right]\left[\sum_{l}^{m}(x_{il}x_{jl})\right]\right\} \text{ can be decomposed into four terms}$$

$$E\left[\text{tr}\left(K^2\right)\right] = \frac{1}{m^2}\left[\sum_{i}^{n}\sum_{k}^{m}x_{ik}^4 + \sum_{i}^{n}\sum_{k\neq l}^{m}x_{ik}^2x_{il}^2 + \sum_{i\neq j}^{n}\sum_{k}^{m}x_{ik}^2x_{jk}^2 + \sum_{i\neq j}^{n}\sum_{k\neq l}^{m}x_{ik}x_{jk}x_{il}x_{jl}\right] \text{ upon } i=j \text{ (or } i\neq j\text{) and } k=l \text{ (or } k\neq l\text{)},$$

and according to Isserlis's Theory [22], having integrated these four terms, we have

$$E[\text{tr}\left(K^2\right)] = \frac{1}{m^2}\left[3nm + n\sum_{k\neq l}^{m}\left(1+2\rho_{kl}^2\right) + n(n-1)m + n(n-1)\sum_{k\neq l}^{m}\rho_{kl}^2\right] = n(n+1)\frac{\sum_{k,l}^{m}\rho_{kl}^2}{m^2} + n = \frac{n(n+1)}{m_e} + n$$

in which $m_e = \frac{m^2}{\sum_{k,l}^{m}\rho_{kl}^2}$ the effective number of markers and $\rho_{kl}^2$ the squared Pearson's correlation of LD between a pair of SNPs [23]. Often $m_e \leq m$, and $m_e = m$ if all markers are in linkage equilibrium (see the note of **Table 1**). Here, $m_e$ is a population parameter, a summary statistic that encompasses allelic frequencies and linkage disequilibrium of makers. According to Eq 3, $E(L_{2,B}) = \text{tr}\left(K^2\right) = \frac{n(n+1)}{m_e} + n$, we consequently propose a randomization algorithm, which estimates $m_e$ as below

$$\begin{cases} \widehat{m}_e = \frac{n(n+1)}{L_{2,B}-n} \\ var\left(\widehat{m}_e\right) = \frac{2m_e^4}{n^4}\frac{\text{tr}(K^4)}{B} \end{cases} \tag{11}$$

A more detailed estimation procedure for $m_e$ can be found in our recent work [16]. See **Note I** in S1 Text for more details.

### Estimation for $\text{tr}(K^4)$

The benchmark estimation for $\text{tr}(K^4)$ is $\text{tr}\left(K^4\right) = \sum_{i=1}^{n}\lambda_i^4$, a fourth-order summation of the eigenvalues of $X$. However, it is computationally expensive when $X$ is large. There are two alternative choices to estimate $\text{tr}(K^4)$. Method I: $\widehat{\text{tr}(K^4)} = \frac{B}{2}var(L_{2,B})$, and $B$ would affect its precision. Method II: $\widehat{\text{tr}(K^4)} = \frac{1}{B}\sum_{b}^{B}z_b^T K^4 z_b$, which uses the fourth-order randomized estimation in Eq 3. Both Method I and Method II can be realized via Eq 3. As will be shown below, Method II provides more stable estimates than Method I.

### About $\Lambda_1$—high-dimension structure of genetic architecture

For $\Lambda_1$, $\Lambda_1 = \text{tr}\left\{\left[\sum(K-I_n)\right]^2\right\} = \text{tr}\left\{\sum(K-I_n)\sum(K-I_n)\right\}$, in which if $\sum = Kh^2 + I\sigma_e^2$ is replaced by $\sum = yy^T$ because $K$ is too expensive to constructed as aforementioned, we consequently have

$$\Lambda_1 \approx \left\{y^T(K-I)K(K-I)y\hat{h}^2 + y^T(K-I)(K-I)y\hat{\sigma}_e^2\right\}$$
$$\approx [L_{3,0} - 2L_{2,0} + L_{1,0}]\hat{h}^2 + [L_{2,0} - 2L_{1,0} + n]\hat{\sigma}_e^2 = \mathcal{L}_{h^2}\hat{h}^2 + \mathcal{L}_{\sigma_e^2}\hat{\sigma}_e^2 \tag{12}$$

**Table 1. Table for high-order moments for different coding scheme for genotypes.**

| Genotype $x_{i,k}x_{i,l}$ | Coding scheme[1,2,3] | Frequencies for $x_{i,k}x_{i,l}$ $(f_{v_1 v_2})$[4] |
|---|---|---|
| $A_kA_kB_lB_l$ | $\alpha_1\beta_1$ | $f_{1,1} = p_k^2 R_{kl}^2 = p_k^2 p_l^2 + 2p_k p_l D_{kl} + D_{kl}^2$ |
| $A_kA_kB_lb_l$ | $\alpha_1\beta_2$ | $f_{1,2} = p_k^2 \cdot 2R_{kl}\overline{R}_{kl} = 2p_k^2 p_l q_l + 2p_k (p_l - q_l) D_{kl} - 2D_{kl}^2$ |
| $A_kA_kb_lb_l$ | $\alpha_1\beta_3$ | $f_{1,3} = p_k^2 \overline{R}_{kl}^2 = p_k^2 q_l^2 - 2p_k q_l D_{kl} + D_{kl}^2$ |
| $A_ka_kB_lB_l$ | $\alpha_2\beta_1$ | $f_{2,1} = 2p_k q_k R_{kl}\bar{r}_{kl} = 2p_k q_k p_l^2 + 2p_l (p_k - q_k) D_{kl} - 2D_{kl}^2$ |
| $A_ka_kB_lb_l$ | $\alpha_2\beta_2$ | $f_{2,2} = 2p_k q_k (\overline{R}_{kl}\bar{r}_{kl} + R_{kl}r_{kl}) = 4p_k q_k p_l q_l + 2(p_k - q_k)(p_l - q_l) D_{kl} + 4D_{kl}^2$ |
| $A_ka_kb_lb_l$ | $\alpha_2\beta_3$ | $f_{2,3} = 2p_k q_k \overline{R}_{kl}r_{kl} = 2p_k q_k q_l^2 + 2q_l (p_k - q_k) D_{kl} - 2D_{kl}^2$ |
| $a_ka_kB_lB_l$ | $\alpha_3\beta_1$ | $f_{3,1} = q_k^2 \bar{r}_{kl}^2 = q_k^2 p_l^2 - 2q_k p_l D_{kl} + D_{kl}^2$ |
| $a_ka_kB_lb_l$ | $\alpha_3\beta_2$ | $f_{3,2} = 2p_k q_k \bar{r}_{kl}r_{kl} = 2q_k^2 p_l q_l + 2q_k (p_l - q_l) D_{kl} - 2D_{kl}^2$ |
| $a_ka_kb_lb_l$ | $\alpha_3\beta_3$ | $f_{3,3} = q_k^2 r_{kl}^2 = q_k^2 q_l^2 + 2q_k q_l D_{kl} + D_{kl}^2$ |

[1]For additive effect, under the coding scheme of 0 ($aa$), 1 ($Aa$), and 2 ($AA$) that counts the number of reference allele ($A$), which has allele frequency of $p$; $q = 1 - p$ is the frequency of the alternative allele. After standardizing each genotype, we have $[\alpha_1, \alpha_2, \alpha_3] = [\frac{2q_k}{\sqrt{2p_k q_k}}, \frac{q_k - p_k}{\sqrt{2p_k q_k}}, \frac{-2p_k}{\sqrt{2p_k q_k}}]$ for $AA$, $Aa$, and $aa$, and $[\beta_1, \beta_2, \beta_3] = [\frac{2q_l}{\sqrt{2p_l q_l}}, \frac{q_l - p_l}{\sqrt{2p_l q_l}}, \frac{-2p_l}{\sqrt{2p_l q_l}}]$ for $BB$, $Bb$, and $bb$. It leads to $\sum_{v_1,v_2}^3 f_{v_1 v_2}\alpha_{v_1}\beta_{v_2} = \frac{D_{kl}}{\sqrt{2p_k q_k 2p_l q_l}} = \rho_{kl}$, in which the subscript $v$ indexes for the three genotypes of a locus.

[2]For dominance effect, under the coding scheme of 0 ($aa$), $2p_l$ ($Aa$), and $4p_l - 2$ ($AA$) for 0, 1, and 2 reference alleles, we have $[\alpha_1, \alpha_2, \alpha_3] = [\frac{-2q_k^2}{\sqrt{4p_k^2 q_k^2}}, \frac{2p_k q_k}{\sqrt{4p_k^2 q_k^2}}, \frac{-2p_k^2}{\sqrt{4p_k^2 q_k^2}},]$ for $AA$, $Aa$, and $aa$, and $[\beta_1, \beta_2, \beta_3] = [\frac{-2q_l^2}{\sqrt{4p_l^2 q_l^2}}, \frac{2p_l q_l}{\sqrt{4p_l^2 q_l^2}}, \frac{-2p_l^2}{\sqrt{4p_l^2 q_l^2}}]$ for $BB$, $Bb$, and $bb$. It leads to $\sum_{v_1,v_2}^3 f_{v_1 v_2}\alpha_{v_1}\beta_{v_2} = \frac{4D_{kl}^2}{\sqrt{4p_k^2 q_k^2 \cdot 4p_l^2 q_l^2}} = \rho_{kl}^2$.

[3]For alternative dominance coding scheme of 0, 1, and 0 for 0, 1, and 2 reference alleles, we have $[\alpha_1, \alpha_2, \alpha_3] = [-\sqrt{\frac{2p_k q_k}{1-2p_k q_k}}, \sqrt{\frac{1-2p_k q_k}{2p_k q_k}}, -\sqrt{\frac{2p_k q_k}{1-2p_k q_k}}]$ and $[\beta_1, \beta_2, \beta_3] = [-\sqrt{\frac{2p_l q_l}{1-2p_l q_l}}, \sqrt{\frac{1-2p_l q_l}{2p_l q_l}}, -\sqrt{\frac{2p_l q_l}{1-2p_l q_l}}]$. It leads to $\sum_{v_1,v_2}^3 f_{v_1 v_2}\alpha_{v_1}\beta_{v_2} = \rho_{kl}\frac{(p_k-q_k)(p_l-q_l)}{\sqrt{(1-2p_k q_k)(1-2p_l q_l)}} + \rho_{kl}^2\sqrt{\frac{2p_l q_l \cdot 2p_k q_k}{(1-2p_k q_k)(1-2p_l q_l)}}$.

[4]The four elements $r_{kl} = q_l + \frac{D_{kl}}{q_k}$, $\bar{r}_{kl} = p_l - \frac{D_{kl}}{q_k}$, $\overline{R}_{kl} = q_l - \frac{D_{kl}}{p_k}$, and $R_{kl} = p_l + \frac{D_{kl}}{p_k}$ represent for conditional probabilities for the four haplotypes $a_k b_l$, $a_k B_l$, $A_k b_l$, and $A_k B_l$, respectively. See **Note V in** S1 Text for detailed calculation.

$L_{3,0} = \boldsymbol{y}^T \boldsymbol{K}^3 \boldsymbol{y}$ can be estimated as in Eq 3 if $\boldsymbol{z}$ is replaced by $\boldsymbol{y}$ the phenotype itself; it is similarly for $L_{2,0} = \boldsymbol{y}^T \boldsymbol{K}^2 \boldsymbol{y}$ and $L_{1,0} = \boldsymbol{y}^T \boldsymbol{K}\boldsymbol{y}$. They reflect high-dimensional structure between $\boldsymbol{y}$ and $\boldsymbol{X}$. So the sampling variance of $h^2$ is not only related to $h^2$ itself, but is eventually upon the high-order structure between $\boldsymbol{y}$ and $\boldsymbol{X}$. See **Note II** in S1 Text for more discussion about $\Lambda_1$.

## About $\eta$ — the term determines the iteration $B$

We define the ratio $\eta$ as below

$$\eta = \frac{\mathrm{tr}\left(\boldsymbol{K}^4\right) h^4}{\Lambda_1}$$

(13)

in which $\mathrm{tr}\left(\boldsymbol{K}^4\right)$ can be estimated as $\widehat{\mathrm{tr}(\boldsymbol{K}^4)} = L_{4,B} = \frac{1}{B}\sum_b^B \boldsymbol{z}_b^T \boldsymbol{K}^4 \boldsymbol{z}_b$ as above. However, it should be noticed that $h^4$ is a heavy penalty for higher heritability; for example, comparing with $h^2 = 0.01$, $h^2 = 0.1$ leads to a 100-fold penalty for the latter in the numerator of Eq 13. Easily, we can estimate $B$ if we want to know how many iterations are needed to reach the preset ratio of $\eta_0$

$$B = \frac{\eta}{\eta_0} = \frac{1}{\eta_0}\frac{\mathrm{tr}\left(\boldsymbol{K}^4\right) h^4}{\Lambda_1}$$

(14)

In practice, $\eta_0$ can take the value of 0.1 or 0.05 as in our simulation and real data analysis below.

## Extended utilities for distributed GWAS datasets

Because datasets are often distributed across institutes, we consequently consider two scenarios for the application of RHE-reg in distributed datasets. As the estimation for $h^2$ (Eq 4) can be split into the numerator and the denominator, the numerator and the denominator are estimated from two different sources. In the other scenario, the whole dataset has been distributed into small slices at $s$ different institutes. We call the first scenario the vertical RHE-reg and the latter horizontal RHE-reg.

### Vertical RHE-reg

Estimation for $h^2$ can be implemented in summary statistics that the numerator and the denominator can be from different components [18]. We denote the correspondingly heritability $\widetilde{h}^2$ for this subtle difference, as well as all tilded symbols from a reference panel that is related to genotypes. Alternatively, Eq 4 can be rewritten as

$$\widehat{\widetilde{h}}^2 = \frac{\widetilde{n}^2(\boldsymbol{y}^T\boldsymbol{K}\boldsymbol{y}-n)}{n^2(\widetilde{L}_{2,B}-\widetilde{n})} = \widetilde{m}_e \cdot \frac{(\boldsymbol{y}^T\boldsymbol{K}\boldsymbol{y}-n)}{n^2} \tag{15}$$

the denominator $\widetilde{L}_{2,B} = \frac{1}{B}\sum_b^B \boldsymbol{z}_b^T \widetilde{\boldsymbol{K}}^2 \boldsymbol{z}_b$. $\widetilde{\boldsymbol{K}} = \frac{1}{m}\widetilde{\boldsymbol{X}}\widetilde{\boldsymbol{X}}^T$, in which $\widetilde{\boldsymbol{X}}$ has the dimension of $\widetilde{n} \times m$; $\widetilde{\boldsymbol{X}}$ is the genotype matrix of the reference sample that is employed to estimate $\widetilde{L}_{2,B}$. So, $var(\widehat{\widetilde{h}}^2)$ is (assuming $n \approx \widetilde{n}$)

$$var\left(\widehat{\widetilde{h}}^2\right) = \frac{2}{\left[\operatorname{tr}\left(\widetilde{\boldsymbol{K}}^2\right)-\widetilde{n}\right]^2} \cdot \left\{\Lambda_1 + \frac{\left[\operatorname{tr}\left(\boldsymbol{K}^2\right)-n\right]^2}{\left[\operatorname{tr}\left(\widetilde{\boldsymbol{K}}^2\right)-\widetilde{n}\right]^2} \cdot \widetilde{h}^4 \cdot \frac{\operatorname{tr}\left(\widetilde{\boldsymbol{K}}^4\right)}{B}\right\} = 2\left(\frac{\widetilde{m}_e}{\widetilde{n}^2}\right)^2 \cdot \left\{\Lambda_1 + \left(\frac{\widetilde{m}_e}{m_e}\right)^2 \cdot \widetilde{h}^4 \cdot \frac{\operatorname{tr}\left(\widetilde{\boldsymbol{K}}^4\right)}{B}\right\}$$

The bias is $E\left[\widehat{\widetilde{h}}^2\right] = h^2 + 2 \cdot \left(\frac{\widetilde{m}_e}{\widetilde{n}^2}\right)^2 \cdot \frac{\operatorname{tr}(\widetilde{\boldsymbol{K}}^4)}{B} \cdot h^2$, which will be zeroed out when $B$ increases (see **Note III in** S1 Text for more general situations). The corresponding test statistic is

$$\widetilde{z}_1 = \left(\frac{\widetilde{n}^2}{\sqrt{2\widetilde{m}_e}}\right) \frac{\widetilde{h}^2}{\sqrt{\Lambda_1}\sqrt{1+\left(\frac{\widetilde{m}_e}{m_e}\right)^2 \frac{\widetilde{\eta}}{B}}}$$

in which $\widetilde{\eta} = \frac{\operatorname{tr}\left(\widetilde{\boldsymbol{K}}^4\right)\widetilde{h}^4}{\Lambda_1}$. For the population of similar ancestry, the ratio $\frac{\widetilde{m}_e}{m_e} \approx 1$ is cancelled out after sufficient iteration, and leads to

$$\widetilde{z}_2 = \left(\frac{\widetilde{n}^2}{\sqrt{2\widetilde{m}_e}}\right) \frac{\widetilde{h}^2}{\sqrt{\Lambda_1}}$$

### Horizontal RHE-reg

For this application, it is assumed that the entire dataset is divided into $s$ institutes ($v$ is subscript for $\boldsymbol{y}$ and $\boldsymbol{X}$). Consequently, $\boldsymbol{y}^T = \left[\boldsymbol{y}_1^T \vdots \boldsymbol{y}_2^T \vdots \cdots \vdots \boldsymbol{y}_s^T\right]$, the whole data $\boldsymbol{y}$ and $\boldsymbol{X}$ are distributed in $s$ institutes, and the length of $\boldsymbol{y}_v$ upon how the proportion of data has in the $v^{th}$ institute; $\boldsymbol{X}^T = \left[\boldsymbol{X}_1^T \vdots \boldsymbol{X}_2^T \vdots \cdots \vdots \boldsymbol{X}_s^T\right]$, similarly the dimension of $\boldsymbol{X}_v$ is $n_v \times m$ in which $n_v$ is the number of individuals in the $v^{th}$ institute. One only needs to receive the mean and summation of square for each $\boldsymbol{y}_v$, and similarly for receiving the allele frequencies of the $m$ reference alleles of $\boldsymbol{X}_v$. So after scaling for $\boldsymbol{y}_v$ and $\boldsymbol{X}_v$,

$$h^2 = \frac{\left\| \left[ \boldsymbol{y}_1^T \vdots \boldsymbol{y}_2^T \vdots \cdots \vdots \boldsymbol{y}_s^T \right]^T \left[ \boldsymbol{X}_1^T \vdots \boldsymbol{X}_2^T \vdots \cdots \vdots \boldsymbol{X}_s^T \right] \right\|_F^2 - n}{\frac{1}{m^2} \left\| \left[ \boldsymbol{Z}_1^T \vdots \boldsymbol{Z}_2^T \vdots \cdots \vdots \boldsymbol{Z}_s^T \right]^T \left[ \boldsymbol{X}_1^T \vdots \boldsymbol{X}_2^T \vdots \cdots \vdots \boldsymbol{X}_s^T \right] \left[ \boldsymbol{X}_1^T \vdots \boldsymbol{X}_2^T \vdots \cdots \vdots \boldsymbol{X}_s^T \right]^T \right\|_F^2 - n} = \frac{\left\| \sum_{v=1}^s \boldsymbol{y}_v^T \boldsymbol{X}_v \right\|_F^2 - n}{\frac{1}{m^2} \left\| \sum_{v=1}^s \boldsymbol{Z}_v^T \boldsymbol{X}_v \boldsymbol{X}_v^T \right\|_F^2 - n}$$

(16)

$\boldsymbol{Z}_v$, a $B \times n_v$ matrix, can be generated from $N(0, 1)$, by each institute, and consequently independently generate $\boldsymbol{y}_v^T \boldsymbol{X}_v$ and $\boldsymbol{Z}_v^T \boldsymbol{X}_v \boldsymbol{X}_v^T$ without compromise of privacy; the subscript $F$ indicates Frobenius norm of a matrix that $\|\boldsymbol{A}\|_F^2 = \left( \sqrt{\operatorname{tr}\left( \boldsymbol{A}\boldsymbol{A}^T \right)} \right)^2 = \operatorname{tr}\left( \boldsymbol{A}\boldsymbol{A}^T \right)$. Upon the precision requirement, after $B$ rounds of iterations, $\eta$ can be calculated so as to evaluate whether further iterations are needed. Unlike the vertical RHE-reg, the horizontal RHE-reg is identical to the RHE-reg under this simple scenario. An R script is attached for its detailed implementation (S1 Data).

### Summary for RHE-reg

Now we discuss some computational issues about RHE-reg. So, eventually $B$ will creep into the RHE-reg. The focus here is to investigate how $B$ would affect the RHE-reg, in particular the stability of $h^2$ and z scores. All the above analyses are based on three computational units, $\boldsymbol{y}$, $\boldsymbol{X}$, and $\boldsymbol{W}$ – if covariates are taken into account, and the operation between them lead to the whole computational procedure, of which their elementary operations can be implemented hierarchically (Table 2). We give an atlas for the computational route. Furthermore, we have $w$ covariates, and the covariate matrix $\boldsymbol{W}$ is of $n \times w$ dimensions. After inclusion of the covariates, the equations for stopping rules can be updated accordingly (see **Note IV in** S1 Text). We finish the description of the statistical approaches and go to their applications now.

**Table 2. Analytical results for RHE-reg.**

| | Individual-level data | | | Vertical RHE-reg | | |
|---|---|---|---|---|---|---|
| $h^2$ estimation | $\begin{cases} h^2 = \frac{\boldsymbol{y}^T \boldsymbol{K}\boldsymbol{y} - n}{L_{2,B} - n} \\ var(\widehat{h^2}) = 2\left(\frac{m_e}{n^2}\right)^2 \left( \Lambda_1 + \frac{\operatorname{tr}(\boldsymbol{K}^4)}{B} h^4 \right) \end{cases}$ | | $z \begin{cases} z_1 = \left(\frac{n^2}{\sqrt{2}m_e}\right)\frac{\widehat{h^2}}{\sqrt{\Lambda_1}\sqrt{1+\frac{\eta}{B}}} \\ z_2 = \left(\frac{n^2}{\sqrt{2}m_e}\right)\frac{\widehat{h^2}}{\sqrt{\Lambda_1}} \end{cases}$ | $\begin{cases} \widetilde{h^2} = \frac{\boldsymbol{y}^T \boldsymbol{K}\boldsymbol{y} - n}{L_{2,B} - \widetilde{n}} \\ var(\widetilde{h^2}) = 2\left(\frac{\widetilde{m_e}}{n^2}\right)^2 \left\{ \Lambda_1 + \left(\frac{\widetilde{m_e}}{m_e}\right)^2 \frac{\operatorname{tr}(\widetilde{\boldsymbol{K}}^4)}{B} h^4 \right\} \end{cases}$ | | $z \begin{cases} \widetilde{z}_1 = \left(\frac{n^2}{\sqrt{2}m_e}\right)\frac{\widetilde{h^2}}{\sqrt{\Lambda_1}\sqrt{1+\left(\frac{\widetilde{m_e}}{m_e}\right)^2\frac{\widetilde{\eta}}{B}}} \\ \widetilde{z}_2 = \left(\frac{n^2}{\sqrt{2}m_e}\right)\frac{\widetilde{h^2}}{\sqrt{\Lambda_1}} \end{cases}$ |
| Key statistics estimation | Intermediate parameters $\begin{cases} \Lambda_1 = \mathcal{L}_{h^2}h^2 + \mathcal{L}_{\sigma_e^2}\sigma_e^2 \\ \eta = \frac{\operatorname{tr}(\boldsymbol{K}^4)\cdot h^4}{\Lambda_1} \end{cases}$ | | $Randomization \begin{cases} \operatorname{tr}\left(\boldsymbol{K}^2\right) = L_{2,B} = \frac{1}{B}\sum_b^B \boldsymbol{z}_b^T \boldsymbol{K}^2 \boldsymbol{z}_b \\ \operatorname{tr}\left(\boldsymbol{K}^4\right) = L_{4,B} = \frac{1}{B}\sum_b^B \boldsymbol{z}_b^T \boldsymbol{K}^4 \boldsymbol{z}_b \\ \widehat{m_e} \begin{cases} \widehat{m_e} = \frac{n(n+1)}{L_{2,B}-n} \\ var\left(\widehat{m_e}\right) = 2\left(\frac{m_e}{n}\right)^4 \cdot \frac{\operatorname{tr}(\boldsymbol{K}^4)}{B} \end{cases} \end{cases}$ | Intermediate parameters $\begin{cases} \widetilde{\Lambda}_1 = \mathcal{L}_{h^2}\widetilde{h^2} + \mathcal{L}_{\sigma_e^2}\widetilde{\sigma}_e^2, \ \widetilde{\eta} = \frac{\operatorname{tr}(\widetilde{\boldsymbol{K}}^4)\cdot h^4}{\Lambda_1} \end{cases}$ | | $Randomization \begin{cases} \operatorname{tr}\left(\widetilde{\boldsymbol{K}}^2\right) = L_{2,B} = \frac{1}{B}\sum_b^B \boldsymbol{z}_b^T \widetilde{\boldsymbol{K}}^2 \boldsymbol{z}_b \\ \operatorname{tr}\left(\widetilde{\boldsymbol{K}}^4\right) = L_{4,B} = \frac{1}{B}\sum_b^B \boldsymbol{z}_b^T \widetilde{\boldsymbol{K}}^4 \boldsymbol{z}_b \\ \widetilde{m_e} \begin{cases} \widehat{m_e} = \frac{\widetilde{n}(\widetilde{n}+1)}{L_{2,B}-\widetilde{n}} \\ var\left(\widehat{m_e}\right) = 2\left(\frac{\widetilde{m_e}}{n}\right)^4 \cdot \frac{\operatorname{tr}(\widetilde{\boldsymbol{K}}^4)}{B} \end{cases} \end{cases}$ |
| | $\begin{aligned} \widehat{\mathcal{L}}_{h^2} &= \frac{1}{m^3}\boldsymbol{y}^T \boldsymbol{K}^3 \boldsymbol{y} - \frac{2}{m^2}\boldsymbol{y}^T \boldsymbol{K}^2 \boldsymbol{y} + \frac{1}{m}\boldsymbol{y}^T \boldsymbol{K}\boldsymbol{y} \\ \widehat{\mathcal{L}}_{\sigma_e^2} &= \frac{1}{m^2}\boldsymbol{y}^T \boldsymbol{K}^2 \boldsymbol{y} - \frac{2}{m}\boldsymbol{y}^T \boldsymbol{K}\boldsymbol{y} + n \end{aligned}$ | | | | | |

**Notes**: left and right parts of the table give how to implement RHE-reg directly in individual-level data or vertical RHE-reg. In order to show the difference between individual-level data estimation and vertical RHE-reg, tilded symbols are introduced to indicate any genotypes from a reference panel. For example, $\widetilde{L}_{2,B} = \sum_{b=1}^B \boldsymbol{z}_b^T \widetilde{\boldsymbol{K}}\widetilde{\boldsymbol{K}}^T \boldsymbol{z}_b$, in which $\boldsymbol{K} = \frac{1}{m}\widetilde{\boldsymbol{X}}\widetilde{\boldsymbol{X}}^T$, and $\widetilde{\boldsymbol{X}}$ is from the reference panel of dimension $\widetilde{n} \times m$.

## Software

We have developed computer software that can handle biobank-scale algorithm presented in this study. The software reads genotype in binary format as defined in such as PLINK. For fast vector-matrix multiplication, Mailman algorithm is employed here [24]. We adapt the implementation of the Mailman algorithm from Agrawal's fast PCA project [25]. It is known that using the Mailman algorithm the vector-matrix multiplication in $L_{2,B}$ is reduced from $\mathcal{O}(nmB)$ to $\mathcal{O}\left(\frac{nmB}{\max(\log_3 n, \log_3 m)}\right)$. There is no conceptual obstacle to applying the method for genotype data in dosage format, but the Mailman algorithm cannot proceed in such a scenario. There are many matrix multiplication included, and in programming some suggested tips to take them out is as shown (S1 Fig).

## Results

### Simulation results

We conducted simulations to evaluate the aforementioned theoretical results under various parameters. The reference allele frequency was evenly sampled from 0.1~0.5, and $h^2$ was set three values of 0, 0.1, and 0.25, and all SNPs were considered causal after a typical polygenic model, which follows Normal distribution. 1) The linkage disequilibrium (Lewontin's $D'$) for each pair of consecutive SNPs were $D' = 0$, 0.2, 0.4, 0.6, and 0.8 for consecutive SNPs. 2) We set three levels of unrelated samples $n = 1,000$, 5,000, and 10,000, respectively. 3) Three levels of SNP numbers $m = 10,000$, 50,000, and 100,000. These five parameters could totally carry out 45 simulation scenarios for each $h^2$ by our in-house simulation code, and its detailed implementation can be found in Zhang et al [16]. For each simulation scenario, we set $B$ the value of 10, 20, and 50 in order to find proper $B$. $n$, $m$ (as well as their allele frequencies), $D'$, and $h^2$ were considered to investigate how to determine $B$. Although neither $n$ nor $m$ reaches real biobank-scale data, we investigate and summarize certain properties of RHE-reg under these 135 scenarios in the results below. The biobank-scale test is to be investigated in UK Biobank examples.

### Result 1: Randomized estimation for tr($K^4$)

As shown in the method section, tr($K^4$) was appeared as one of the key parameters in determining the performance of the sampling of RHE-reg. The direct estimation of $\text{tr}\left(K^4\right)$ from the eigenvalues of $K$ was the golden standard, and we consequently compared Method I, $\widehat{\text{tr}(K^4)} = \frac{B}{2} var(L_{2,B})$, and Method II, $\widehat{\text{tr}(K^4)} = L_{4,B}$, with its direct estimation. As shown in Fig 1, the above 135 simulation scenarios were compared with the direct estimation for $\text{tr}\left(K^4\right) = \sum_{i=1}^{n} \lambda_i^4$. For Method I, increasing $B$ from 10 to 50 could increase the precision of the estimation. In contrast, Method II showed very consistent and high precision for the estimation of tr($K^4$) regardless of the sample size, an increasing of $B$ from 10 to 50 did not help improve precision. The advantage of Method II was probably because $L_{4,B}$ estimated tr($K^4$) as its mean, whereas $var(L_{2,y})$ as its sampling variance. So, hereafter we used Method II $L_{4,B}$ to estimate tr($K^4$). Of note, as $\text{tr}\left(K^4\right) = \sum_{i=1}^{n} \lambda_i^4$ is computational expensive when $K$ is large so that only limited sample size and SNP numbers were tested in these 135 simulations; however, the principal results should be retained for an even larger sample size, as well as $K$, but with more expensive computational cost in solving eigenvalues. Biobank-scale performance of the proposed method will be illustrated in UK Biobank 81 traits in **Result 4**.

### Result 2: MSE of RHE-reg

In Eq 7, $MSE\left(\widehat{h^2}\right) = 2\left(\frac{m_e}{n^2}\right)^2\left(\Lambda_1 + \frac{\text{tr}(K^4) \cdot h^4}{B}\right) + 4\left(\frac{m_e}{n^2}\right)^4 \cdot \left[\frac{\text{tr}(K^4)}{B}\right]^2 \cdot h^4$, and we defined $R = \frac{\frac{\text{tr}(K^4) \cdot h^4}{B}}{\Lambda_1} = \frac{\eta}{B}$ according to Eq 14. In Fig 2, we showed how MSE and $R$ could be reduced by $B$ for these 90 simulated scenarios (excluded 45 scenarios under $n = 1,000$). We only illustrated the results for $n = 5,000$ and 10,000, respectively, because $n = 1,000$ was too small a sample size here for efficient convergence. The top row of Fig 2 illustrated how MSE were reduced by $B$, and obviously

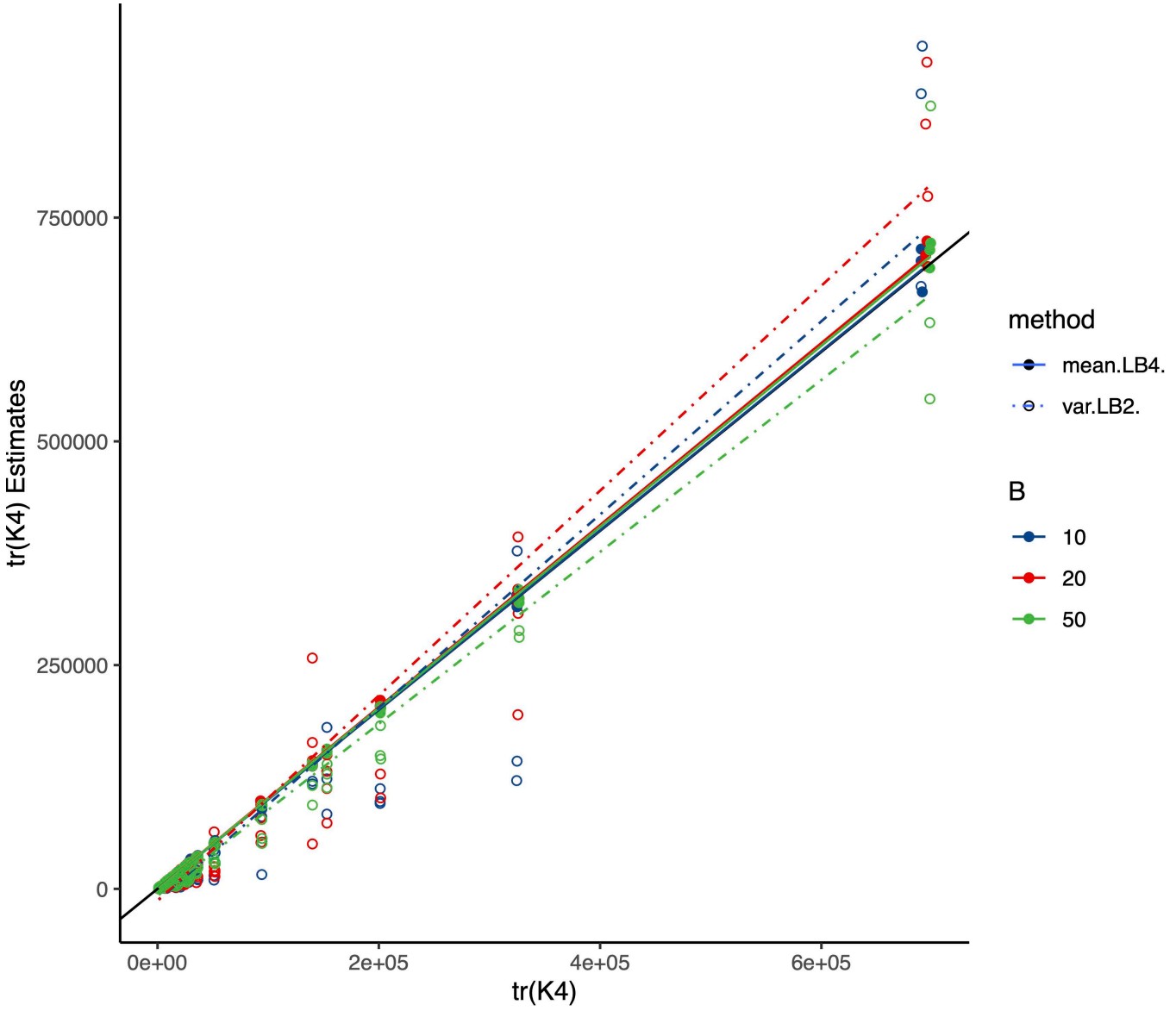

**Fig 1. Comparison for the estimation of** $\text{tr}(\boldsymbol{K}^4)$**.** The x-axis represents benchmark estimation for $\text{tr}(\boldsymbol{K}^4)$ directly, and y-axis represents the estimation of $\text{tr}(\boldsymbol{K}^4)$ using Method I or Method II respectively. The diagonal line (solid black) is for comparison. Each fitted line shows the correlation between all 135 estimations with their benchmark estimation $\text{tr}(\boldsymbol{K}^4)$.

a much larger $B$ reduced MSE because $\frac{\text{tr}(\boldsymbol{K}^4) \cdot h^4}{B}$ was turned down. Actually the bias term $4\left(\frac{m_e}{n^2}\right)^4 \cdot \left[\frac{\text{tr}(\boldsymbol{K}^4)}{B}\right]^2 \cdot h^4$ played little weight in MSE, which was dominantly determined by $2\left(\frac{m_e}{n^2}\right)^2\left(\Lambda_1 + \frac{\text{tr}(\boldsymbol{K}^4)\cdot h^4}{B}\right)$. $2\left(\frac{m_e}{n^2}\right)^2\left(\Lambda_1 + \frac{\text{tr}(\boldsymbol{K}^4)\cdot h^4}{B}\right)$ was at least one or two order of magnitude compared with $4\left(\frac{m_e}{n^2}\right)^4 \cdot \left[\frac{\text{tr}(\boldsymbol{K}^4)}{B}\right]^2 \cdot h^4$. In the second row of Fig 2 $R = \eta/B$ reflected how quickly $\frac{\text{tr}(\boldsymbol{K}^4)\cdot h^4}{B}$ vanished after $B$ iterations. Neither LD nor $h^2$ played an important role in determining MSE for the simulated scenarios, but the ratio between $n$ and $m$ mattered much as under the same sample size, more SNPs always inflated MSE.

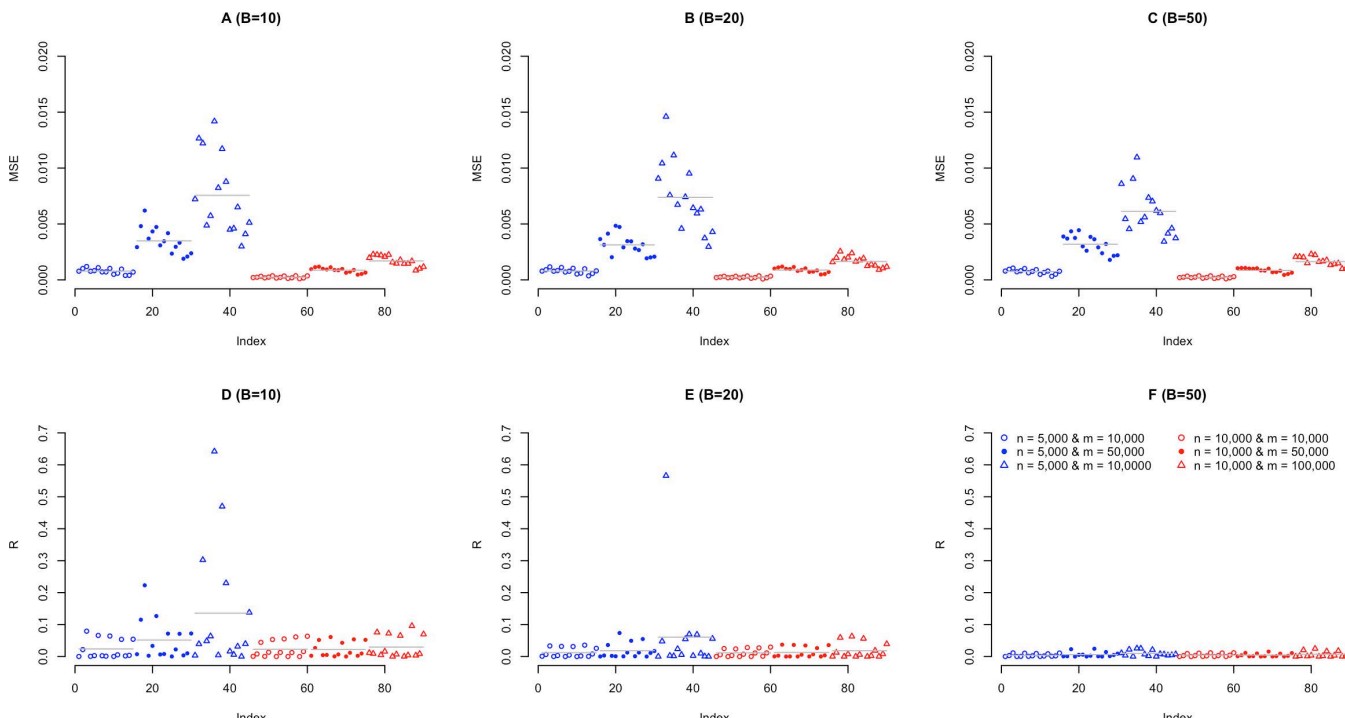

**Fig 2. Evaluation for the MSE of RHE-reg under the different simulation scenarios.** The top row (**A-C**) represents the comparison for MSE under different $B$ for 90 simulated scenarios, and the bottom row (**D-F**) represents the comparison for the ratio between $\Lambda_1$ and $\frac{\mathrm{tr}(\boldsymbol{K}^4) \cdot h^4}{B}$. In each panel, 90 simulated scenarios are split into 6 groups given different combination for sample sizes ($n$ = 5,000, and 10,000) and SNP numbers ($m$ = 10,000, 50,000, and 100,000). In each group 15 points can be split into 5 groups from left to right for different LD levels ($D'$ = 0, 0.2, 0.4, 0.6, and 0.8) and each LD group has three simulated $h^2$ (0, 0.1, and 0.25), respectively. In each panel, the grey line indicates the mean of the investigated values for the corresponding 15 scenarios.

## Result 3: Randomized estimation for $h^2$ and z-score

In result 3, we studied how $B$ could influence $h^2$ and its $z$-score. As the sampling variance of $h^2$ was reciprocal to the sample size $\Lambda_0 = \frac{2m_e}{n^2}$ under the null hypothesis and $B$, it was obviously to see in simulation that: greater $n$, and greater $B$ would help to bring out a more stable estimation for $h^2$ (Fig 3A-C). If we employed $\hat{h}^2$ from $B = 50$ as the benchmark, when sample size $n = 10,000$, there was very high consistent estimation for $\hat{h}^2$ even $B$ $B = 20$ (Fig 3C). $\frac{2m_e}{n^2}$ is the sampling variance of REML when $h^2 = 0$ [26].

The availability of the $z$ score of the estimated heritability was important for statistical inference. We evaluated the influence of $B$ in determining the performance of the randomized algorithm (Fig 3D-F). It was known from the above analysis $E(z_{h^2}) \approx \frac{n^2}{\sqrt{2m_e}} \frac{\hat{h}^2}{\sqrt{\mathcal{L}_{h^2}\hat{h}^2 + \mathcal{L}_{\sigma_e^2}\hat{\sigma}_e^2}}$, so when the estimation of $h^2$ became stable the test statistic was stable too. So, $z_1$ was relative stable when $n = 5,000$ (Fig 3E) or $n = 10,000$ (Fig 3F). When the sample size was sufficiently large, a few iteration could guarantee high accuracy of the estimation. In addition, we also tested the estimation by setting $h^2 = 0.5, 0.75$, and 0.9, respectively, and the results, as promised by our theory, were consistent to what observed.

## Result 4: Application of horizontal RHE-reg

This study was to estimate heritability for distributed data as exact as a single piece of data. Two cohorts with $n_1 = 4,000$ and $n_2 = 6,000$ individuals, respectively, were generated to verify h-RHE-reg. $h^2$ was set the value of 0, 0.1, and

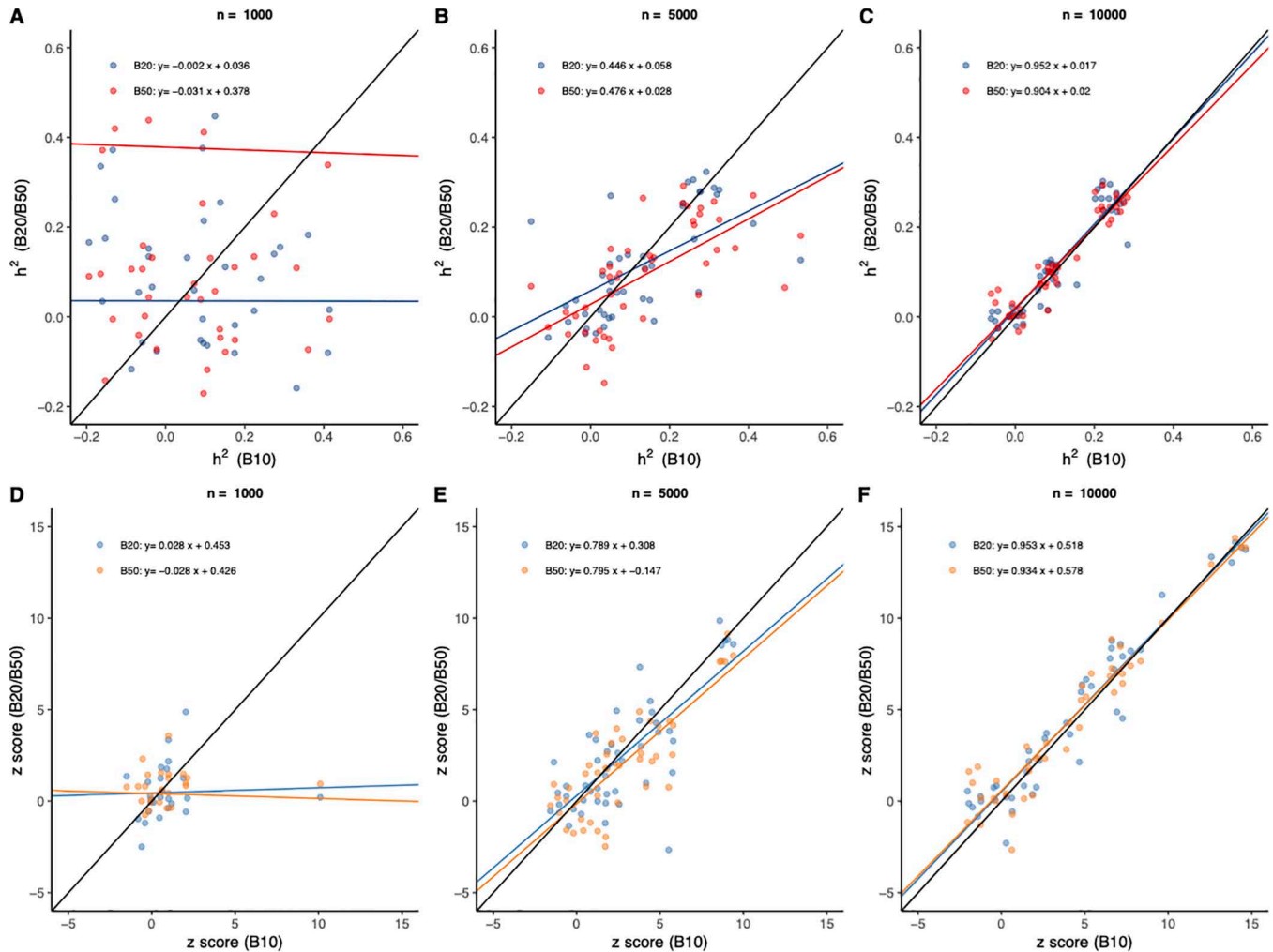

**Fig 3. Estimation of h²and z-score after different B.** Each plot illustrates the comparison of the estimated heritability **(A-C)** and z-score **(D-F)** given B = 10 (x-axis) vs B =20 and 50 (y-axis) under different sample size $n = 1,000$, 5,000, and 10,000, respectively. The black solid line is the reference line of y = x, and the coloured solid line is the fitted regression, which is printed in each plot. In each plot, there are 45 points in each colour as simulated.

0.25, respectively. The effects of all $m = 10,000$ SNPs were sampled from the distribution $N(0, \frac{h^2}{m})$. Heritability and z scores were estimated using individual-level RHE-reg as well as h-RHE-reg. $B$ was set of 10, 20, and 50. The genotypes of the two simulated cohorts were standardized by $\widetilde{x}_{1j} = \frac{x_{1j} - 2\bar{p}_j}{\sqrt{2\bar{p}_j(1-\bar{p}_j)}}$ and $\widetilde{x}_{2j} = \frac{x_{2j} - 2\bar{p}_j}{\sqrt{2\bar{p}_j(1-\bar{p}_j)}}$ for the $j$-th locus, where $\bar{p}_j = \frac{n_1}{n_1+n_2}\bar{p}_{1j} + \frac{n_2}{n_1+n_2}\bar{p}_{2j}$ was the average allele frequency. The phenotypes of the two cohorts were standardized by $\widetilde{y}_1 = \frac{y_1 - \bar{y}}{\sigma_y}$ and $\widetilde{y}_2 = \frac{y_2 - \bar{y}}{\sigma_y}$, where $\bar{y} = \frac{n_1}{n_1+n_2}\bar{y}_1 + \frac{n_2}{n_1+n_2}\bar{y}_2$ and $\sigma_y^2 = \frac{n_1}{n_1+n_2-1}\bar{y_1^2} + \frac{n_2}{n_1+n_2-1}\bar{y_2^2} - \frac{n_1+n_2}{n_1+n_2-1}\bar{y}^2$. Each simulation scenario had 10 repeats (see Source1.R and Source2.R in S1 Data for its implementation).

The estimates of heritability and its z score were consistent using individual-level RHE-reg and h-RHE-reg in all scenarios when the random vectors were the same (Fig 4). We also split the data into $n_1 = 2,000$ and $n_2 = 8,000$ individuals, and, as expected, the results were nearly identical and unbiased.

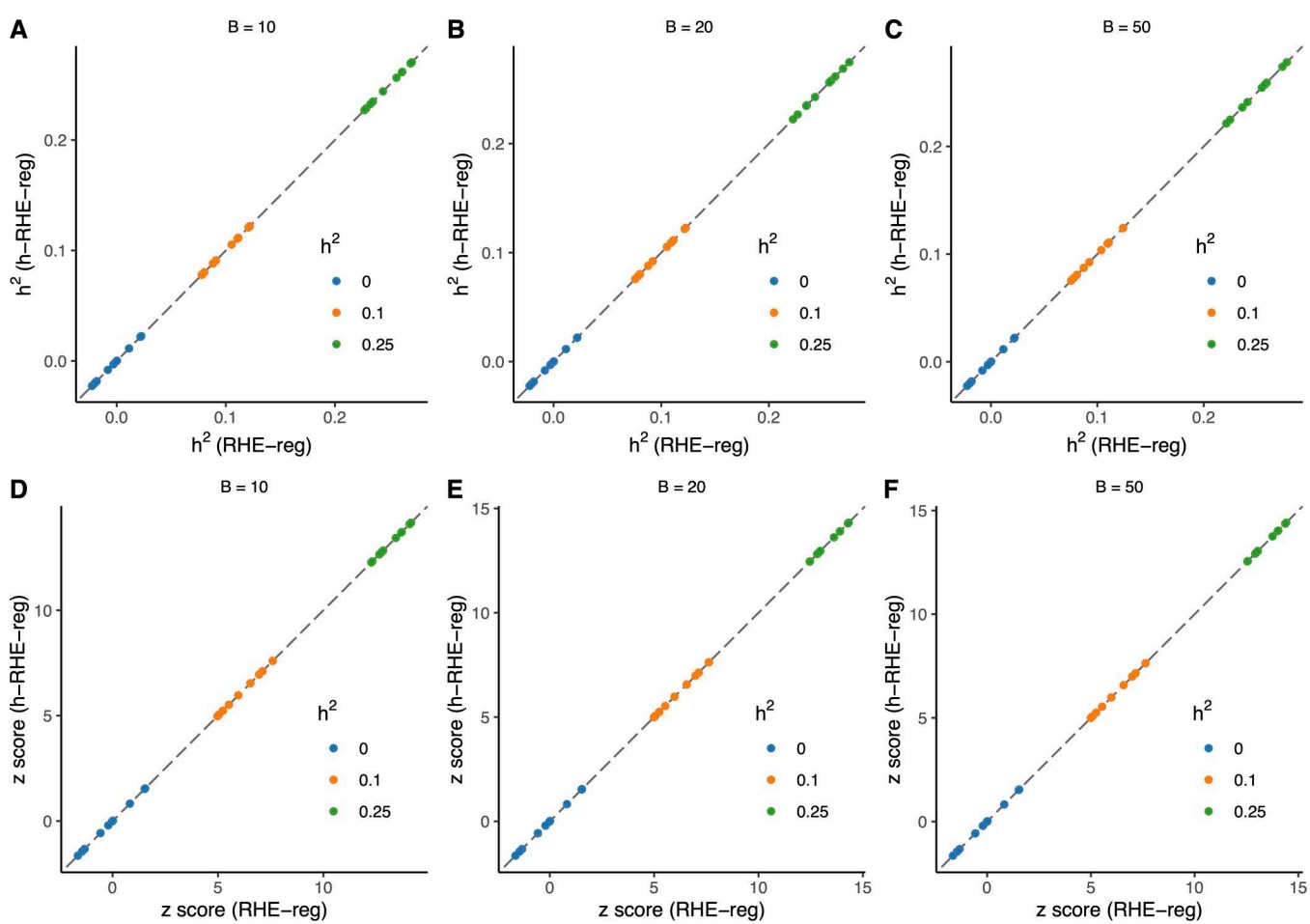

**Fig 4. Application of horizontal RHE-reg in simulation studies.** Estimated heritability (**A-C**) and z-scores (**D-F**) obtained from RHE-reg (x axis) and h-RHE-reg (y axis) under different settings of $B=10$, 20, and 50, respectively. Point colors represent the simulated heritability. Each scenario was repeated 10 times. The dashed line represents the identity line (y = x).

## Real data analysis for UK Biobank

We chose the unrelated 292,223 British white who have no kinship found, as indicated by the genetic kinship provided in the UK Biobank (field 22021) for real data test [2]. After quality control, the inclusion criteria were: MAF > 0.01, missing call rate < 0.05 and Hardy-Weinberg proportion test *p*-value > 1e-6, whose genotype call rate > 0.95, and 525,460 autosome SNPs were included for analysis. We estimated heritability of the 81 quantitative traits, and included the top two principal components and sex as covariates.

We used two strategies to estimate heritability. In strategy I, denoted as $B+$ strategy hereafter, we set $B_0 = 10$ as a warm-up step to evaluate $\text{tr}(\boldsymbol{K}^4)$ and $\eta_0$ was set of 0.05. After the warm-up of $B_0$ iteration, we then increased iteration by a step of 10, We then estimate final realized $\eta$, $m_e$, $h^2$, and three kinds of $z$ scores until the convergence ratio of $\eta_0 = 0.05\,\Lambda$; however, we set a hard stop for $B_1 = 200$ even if $\eta$ was still greater than 0.05. In strategy II, we directly set $B_0 = 10$, 20, or 50 without further considering additional iteration anymore, and consequently denoted as $B10$, $B20$, and $B50$ strategies hereafter.

## Comparison of the UKB results between strategy I and II

Even a couple of traits were set to take a hard stop because their $B_1$ were greater than 200, the estimated $\hat{\eta}$ for the 81 traits had a mean of 0.0518, which was very close to the preset $\eta_0 = 0.05$ (Fig 5A1). It indicated that our theory worked to control the precision of the sampling variance of RHE-reg. The trait "age of diabetes diagnosed" had $h^2 = 1.21 \pm 0.533$, extremely large standard error compared to other traits, because of its smallest sample size of $n = 12,658$ (Fig 5B1). In Fig 5C1, we got three $z$ scores, which are $z_1 = \sqrt{\frac{\hat{\Lambda}_0}{2\hat{\Lambda}_1}}\frac{\hat{h}^2}{\sqrt{1+\frac{\hat{\eta}}{B_1}}}$ score directly calculated given $B_1$ iterations (green colored, via Eq 8), the optimal z score $z_2 = \sqrt{\frac{\hat{\Lambda}_0}{2\hat{\Lambda}_1}}\hat{h}^2$ when $B$ was infinitive (blue colored, via Eq 9), and the predicted $z_3 = \hat{z}\sqrt{1+\frac{\hat{\eta}}{B}}$ score (pink colored, via Eq 10).

For comparison, we examined the corresponding statistics that were estimated under $B10$, $B20$, and $B50$, respectively. In strategy II, A larger $B_0$ led a smaller $\eta$ as expected (Fig 5A2-4). Interestingly, regardless the change of $B_0$ in strategy II, $\hat{h}^2$ were very consistent to those estimated from strategy I, as shown that the fitted regression lines were very close to 1

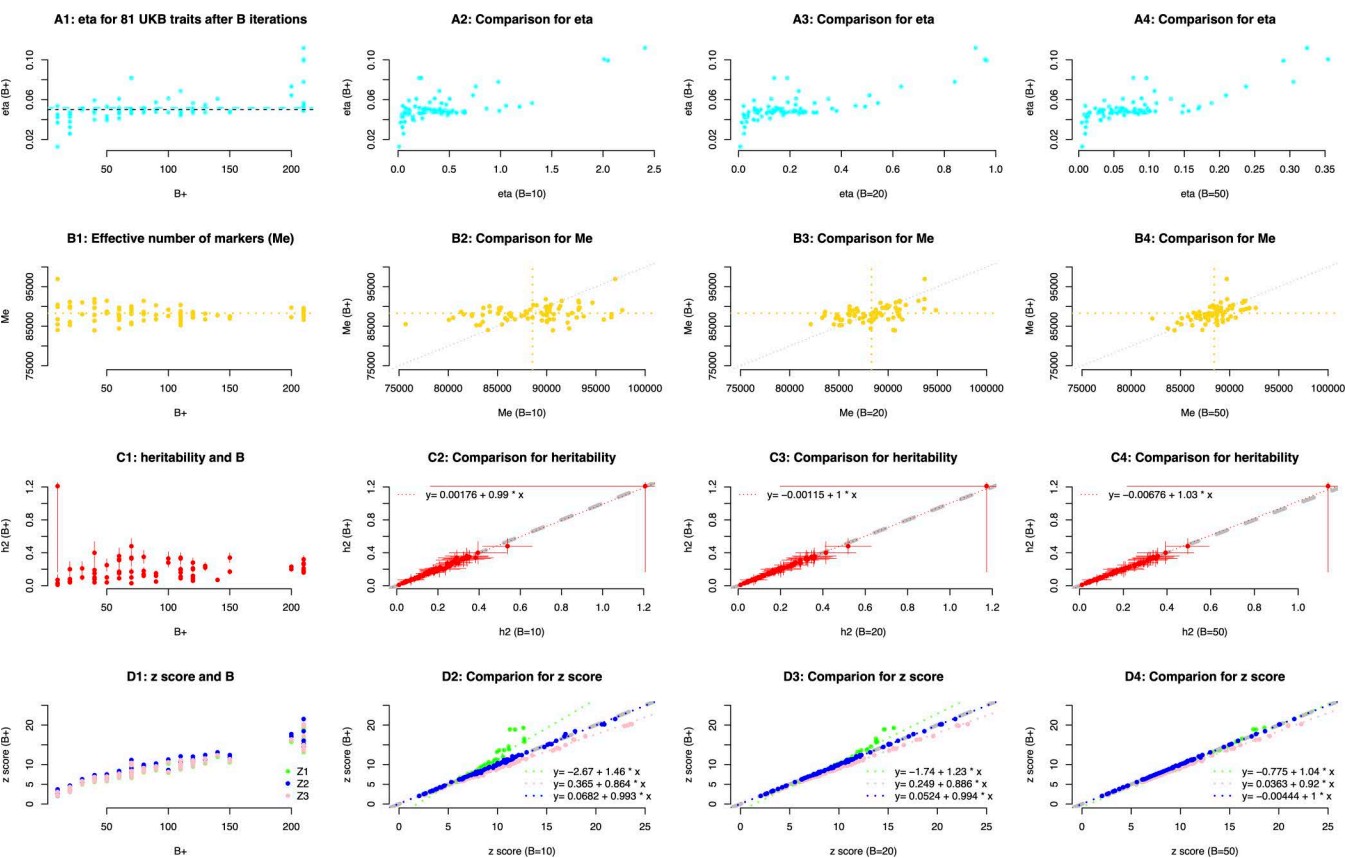

**Fig 5. Randomized estimation for heritability for UK Biobank 81 quantitative traits. A1-D1)** The performance of RHE-reg given the respective $B$ number for each trait, $\eta$, effective number of markers using randomized estimation ($m_e$), $\hat{h}^2$ (the vertical line covers 95% confidence interval), and $z$ scores estimated in three methods. Three z scores are plotted, the green colored z scores are directly estimated given $B$ iterations for each trait (Eq 8), the pink colored $z$ scores are optimal z score (Eq 9), and the blue colored z scores are directly estimated given $z\sqrt{1+\eta}$ (Eq 10). **A2-A4)** Comparison for $\eta$ between that of $B+$ and $B = 10$, 20, and 50, respectively. **B2-B4)** Comparison for $m_e$ between that of $\mathbf{B}+$ and $B = 10$, 20, and 50, respectively; the vertical and horizontal lines are the means of $m_e$ from x-axis and y-axis, respectively. **C2-C4)** Comparison for $\mathbf{h}^2$ between $B+$ and $B = 10$, 20, and 50, respectively; the fitted lines is printed on the top left corner of each plot. **D2-D4)** Comparison for the three pairwise $z$ scores. The green colored z scores are estimated in Eq 8 given B+ and the number of $B$ as shown on the x-axis label, the pink colored $z$ score are estimated in Eq 9, and blue colored ones in Eq 10, respectively.

(Fig 5B2-B4). Three types of *z* scores were compared (Fig 5C2-C4), and the optimal *z* scores from both strategies were nearly perfect (blue points and blue dashed lines). Then, as shown in Fig 5C4, the three kinds of *z* scores were nearly completely matched.

In addition, the estimates were also consistent with our previous results using a less efficient method [27], and see S1 Table for more details. The heritability estimated by the randomization algorithm exhibited a relative high degree of correlation (Pearson's correlation coefficient of 0.77) with the previous estimates for 81 traits. Compared to the previous results, the $m_e$ was nearly consistent with the GRM-based estimates, and is with averaged 1.38% deviation after 10 iterations and further decreased to 1.23% deviation after 50 iterations (S1 Table).

We also compared the computational efficiency of RHE-reg with GCTA [28] and BOLT-REML [29] in estimating the heritability on BMI. The comparison was conducted on a sub-dataset in UKB with randomly selected 10,000 individuals and 523,945 SNP markers after filtration. The results indicate significant efficiency improvement in estimating the heritability of complex traits in biobank-scale datasets for RHE-reg, with computation times reduced by 96.6% and 83.8% compared to GCTA and BOLT-REML, respectively (S2 Table). More benchmark comparison of the computational performance could be found in earlier studies [18,5]. Even using a complete dataset, RHE-reg could also complete heritability estimation within an acceptable time (S3 Table). In our tested 81 UKB traits, with 10 threads, it on average took 453 mins to finish the analysis of a trait and the average iteration of $B = 90$.

## Application of vertical RHE-reg

Of Eq 15, $\hat{\tilde{h}}^2 = \widetilde{m}_e \cdot \frac{(y^T K y - n)}{n^2}$ indicates that $\widetilde{m}_e$ and $\frac{(y^T K y - n)}{n^2}$ can be from two independent sources. Consequently, we split each UKB trait evenly into halves to test the v-RHE-reg, and Eq 15 had four possible combinations: 1) split 1/1: both $\widetilde{m}_e$ and $\frac{(y^T K y - n)}{n^2}$ were estimated from split 1; 2) split 2/1: $\widetilde{m}_e$ was estimated from split 2 and $\frac{(y^T K y - n)}{n^2}$ split 1; 3) split 1/2: $\widetilde{m}_e$ was estimated from split 1 and $\frac{(y^T K y - n)}{n^2}$ split 2; 4) both $\widetilde{m}_e$ and $\frac{(y^T K y - n)}{n^2}$ were estimated from split 2. So, we had four estimators as below

$$
\begin{cases}
h^2_{1,1} = \left[\widetilde{m}_e\right]_1 \cdot \left[\frac{(y^T K y - n)}{n^2}\right]_1, & \text{split } 1/1 \\
h^2_{1,2} = \left[\widetilde{m}_e\right]_1 \cdot \left[\frac{(y^T K y - n)}{n^2}\right]_2, & \text{split } 2/1 \\
h^2_{2,1} = \left[\widetilde{m}_e\right]_2 \cdot \left[\frac{(y^T K y - n)}{n^2}\right]_1, & \text{split } 1/2 \\
h^2_{2,2} = \left[\widetilde{m}_e\right]_2 \cdot \left[\frac{(y^T K y - n)}{n^2}\right]_2, & \text{split } 2/2
\end{cases}
$$

Of each trait, its heritability and *z* score tests could be constructed within each split and between each split by exchanging the $L_B$ estimation, and consequently brought out v-RHE-reg. As shown in Eq 15, we compared the result for *B*=10, 20, and 50, respectively, and observed consistent results between split 1 and split 2, and between split 1/2 and split 2/1.

Fig 6 showed the results of these four estimators under different *B*. It illustrated that pairwise estimates $\hat{h}^2_{1,1}$ against $\hat{h}^2_{2,2}$, and $\hat{h}^2_{1,2}$ against $\hat{h}^2_{2,1}$, and as observed the pairwise estimates were quite consistent with each other both within and between splits.

## Discussion

The presented study is developed on the randomized Haseman-Elston regression for the estimation of SNP-heritability proposed recently by Wu and Sankararaman (2018) [5]. They very smartly used a randomization approach – Girard-Hutchinson estimation, which significantly reduces the computational cost in estimating $\text{tr}(K^2)$ from $\mathcal{O}(n^2 m)$ to $\mathcal{O}(nmB)$ [20,19]. However, the drawbacks of their method may be its unclear property for *B*, which further leads to obscure

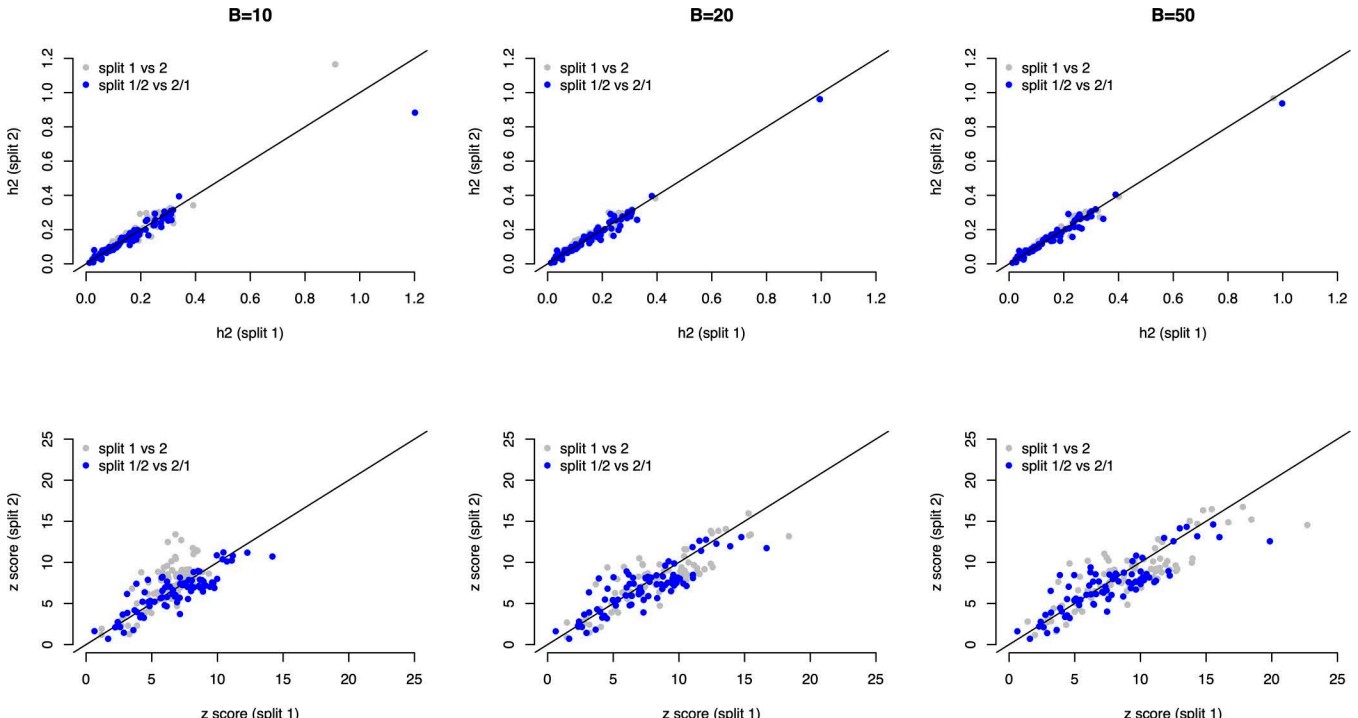

**Fig 6. Application of v-RHE-reg for 81 quantitative traits in UKB.** The top and the bottom row represent heritability and z score respectively. Each column illustrates results after $B$ iterations. In each plot, the coordinate for a grey point is heritability/z-score estimated from split 1 (x-axis) and split 2 (y-axis).

sampling variance of the estimated heritability. As discussed in a recent review, it has been obscure in the original RHE-reg since no closed-form solutions were provided to quantify the connection between $B$ and the estimation procedure [8]. After integrating analytical results for Haseman-Elston regression into this randomized framework [4], we present here a close-form solution for RHE-reg. Having provided the sampling variance, we are able to evaluate how $B$ influences the estimation procedure of RHE-reg precisely. In particular, a key element that is related to the sampling variance of $L_{2,B}$, which is proportional to $\frac{2\text{tr}(\boldsymbol{\kappa}^4)}{B}$. It should be noticed $var(\hat{h}^2) = \frac{2m_e}{n^2}$ under the null hypothesis that $h^2 = 0$ as established previously [4,26,18]. The quantity of $\frac{2m_e}{n^2}$ is identical to the sampling variance of REML under the null hypothesis or that of modified Haseman-Elston regression [26,4,18]. Of note, the present study is focused on the presence of typical polygenic architecture because counterexample, albeit pathological, can be found when causal variants are distributed not random as discussed [4,27].

A nature extension of the method is to include multi-component, such as for the estimation for each chromosome. It is obvious that the method for deriving sampling variance should be extended for multi-components estimation if their corresponding $\boldsymbol{X}_i$ and $\boldsymbol{X}_j$ are in global linkage, or nearly, equilibrium, which is often the case for human populations [17]. Much advanced numerical tools, such as condition numbers, are needed to evaluate the approximation of the randomized algorithm [30]. Some inconsistency between GRM-based estimation and randomization estimation, such as the overall correlation of 0.77 for estimated heritability between Xu et al.'s results and the current result, may arise from the different covariates chosen [27]. In Xu et al.'s work, the heritability was estimated under the first two PCs corrected, while the current randomization method further took gender as extra covariate, this may cause the observed discrepancy especially in the gender-related traits.

In summary, the purpose of the present study is two-fold. First, we provide a method to balance iteration and precision of estimation, and an improved implementation of RHE-reg is realized. Secondly, we extend RHE-reg into the estimation of SNP-heritability for distributed data, which uses the controlled $B$ to synchronize the estimation across datasets. With increasing genomic cohorts but distributed in different institutes, it is now a trend to propose computational solutions without compromising privacy [31]. The enhanced RHE-reg framework can consequently have computational and analytical merits, and, as demonstrated, we further extend its utilities such as vertical- and horizontal RHE-reg, as demonstrated in this study. Given the increasing cry for genomic privacy, both vertical and horizontal RHE-reg will be meaningful in securing genomic information. However, given its traditionally very quantitative origin of statistical genetics, statistical routines may have competing, if not superior, solutions than those derived from available information technology [32,12].

It is straightforward to apply the estimation procedure for the estimation of dominance variance components both for individual-level data and summary statistics. The only update of the equation $h_d^2 = \frac{\boldsymbol{y}^T \boldsymbol{K}_d \boldsymbol{y} - n}{\mathrm{tr}(\boldsymbol{K}_d^2) - n}$ is to replace $\boldsymbol{K}$ with $\boldsymbol{K}_d = \frac{1}{m} \boldsymbol{X}_d \boldsymbol{X}_d^T$. For each SNP, $\boldsymbol{x}_{i,d}$ is coded 0, 2p, and 4p-2 for the genotype that counts 0, 1 and 2 reference alleles; and furthermore, $\boldsymbol{X}_d$ is further scale by $\frac{X_{d,l} - 2p_l^2}{2p_i(1-p_i)}$ [33,34]. So for a pair of individual $i$ and $j$, $\boldsymbol{K}_{d[i,j]} = \frac{1}{m} \Sigma_l^m \frac{(X_{d,i,l} - 2p_l^2)(X_{d,j,l} - 2p_l^2)}{4p_l^2(1-p_i)^2}$. After replacing $\boldsymbol{K}$ with $\boldsymbol{K}_d$, all the above estimation procedure can be applied for $h_d^2$. Furthermore, $\mathrm{tr}\left(\boldsymbol{K}_d^2\right) = n(n+1)\frac{\sum_{k,l}^m \rho_{kl}^4}{m^2} + n = \frac{n(n+1)}{m_{e.d}} + n$. The effective number of markers in terms of $\boldsymbol{X}_d$ is $m_{e.d} = \frac{m^2}{m + \sum_{k \neq l}^m \rho_{kl}^4}$, a tetradic form of LD for a pair of SNPs.

## Supporting information

**S1 Text. Technical details on some mathematical derivations.** "Effective number of markers" (Note I); "Discussion of $\Lambda_1$" (Note II); "Sampling variance for vertical RHE-reg" (Note III); "Adjustment for covariates" (Note IV); "Coding scheme and LD" (Note V).
(DOCX)

**S1 Fig. This figure gives the computational tips on how to program the software.** Some sequential operation of the matrix is suggested to make the program easy to write.
(TIFF)

**S1 Data. Implementation for Horizontal RHE-reg (R code: Source1. R and Source2.R).**
(ZIP)

**S1 Table. SNP-heritability estimation for 81 UKB traits (XLSX).**
(XLSX)

**S2 Table. The time cost for heritability estimation on BMI for RHE-reg, BOLT-REML and GCTA.**
(XLSX)

**S3 Table. Computational time for 81 continuous traits using RHE-reg.**
(XLSX)

## Acknowledgments

We thank the participants of the included cohorts and of UK Biobank for making this work possible (UKB application 41376).

## Author contributions

**Data curation:** Guo-An Qi, Qi-Xin Zhang.

**Formal analysis:** Guo-An Qi, Qi-Xin Zhang, Zhe Fan.

**Funding acquisition:** Zhe Zhang, Siyang Liu, Guo-Bo Chen.

**Investigation:** Guo-An Qi, Qi-Xin Zhang, Xiyun Xu, Zhe Zhang, Siyang Liu, Guo-Bo Chen.

**Methodology:** Jingyu Kang, Tianyuan Li, Guo-Bo Chen.

**Resources:** Guo-Bo Chen.

**Software:** Guo-Bo Chen.

**Supervision:** Siyang Liu, Guo-Bo Chen.

**Validation:** Guo-An Qi, Qi-Xin Zhang, Jingyu Kang, Xiyun Xu, Zhe Zhang.

**Visualization:** Guo-An Qi, Guo-Bo Chen.

**Writing – original draft:** Guo-Bo Chen.

**Writing – review & editing:** Qi-Xin Zhang.

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
