## [Decision Letter · Decision Letter 0]

10 Jun 2025

PCOMPBIOL-D-25-00059

Analytical and computational solution for the estimation of SNP-heritability in biobank-scale and distributed datasets

PLOS Computational Biology

Dear Dr. Chen,

Thank you for submitting your manuscript to PLOS Computational Biology. After careful consideration, we feel that it has merit but does not fully meet PLOS Computational Biology's publication criteria as it currently stands. Therefore, we invite you to submit a revised version of the manuscript that addresses the points raised during the review process.

Please submit your revised manuscript within 60 days Aug 10 2025 11:59PM. If you will need more time than this to complete your revisions, please reply to this message or contact the journal office at ploscompbiol@plos.org. Please include the following items when submitting your revised manuscript:

We look forward to receiving your revised manuscript.

Kind regards,

Androniki Psifidi, DVM, PhD

Guest Editor

PLOS Computational Biology

Ilya Ioshikhes

Section Editor

PLOS Computational Biology

**Journal Requirements:** **1)** Please provide an Author Summary. This should appear in your manuscript between the Abstract (if applicable) and the Introduction, and should be 150-200 words long. The aim should be to make your findings accessible to a wide audience that includes both scientists and non-scientists. Sample summaries can be found on our website under Submission Guidelines:https://journals.plos.org/ploscompbiol/s/submission-guidelines#loc-parts-of-a-submission **2)** Please upload all main figures as separate Figure files in .tif or .eps format. For more information about how to convert and format your figure files please see our guidelines: https://journals.plos.org/ploscompbiol/s/figures **3)** We have noticed that you have uploaded Supporting Information files, but you have not included a list of legends. Please add a full list of legends for your Supporting Information files after the references list. **4)** Please note that your Data Availability Statement is currently missing the repository name, and the DOI/accession number of each dataset OR a direct link to access each dataset. If your manuscript is accepted for publication, you will be asked to provide these details on a very short timeline. We therefore suggest that you provide this information now, though we will not hold up the peer review process if you are unable. **5)** Please provide a detailed Financial Disclosure statement. This is published with the article. It must therefore be completed in full sentences and contain the exact wording you wish to be published.1) Please clarify all sources of financial support for your study. List the grants, grant numbers, and organizations that funded your study, including funding received from your institution. Please note that suppliers of material support, including research materials, should be recognized in the Acknowledgements section rather than in the Financial Disclosure2) State the initials, alongside each funding source, of each author to receive each grant. For example: "This work was supported by the National Institutes of Health (####### to AM; ###### to CJ) and the National Science Foundation (###### to AM)."3) State what role the funders took in the study. If the funders had no role in your study, please state: "The funders had no role in study design, data collection and analysis, decision to publish, or preparation of the manuscript."4) If any authors received a salary from any of your funders, please state which authors and which funders..If you did not receive any funding for this study, please simply state: u201cThe authors received no specific funding for this work.u201d **6)** Your current Financial Disclosure states, "The author(s) received no specific funding for this work.".However, your funding information on the submission form indicates National Natural Science Foundation of China 31771392 to Dr. Guo-Bo Chen, Natural Science Foundation of Jilin Province 32102503to Zhe Zhang, Shenzhen Basic Research Foundation 20220818100717002 to Siyang Liu and Basic and Applied Basic Research Foundation of Guangdong Province 2022B1515120080 to Siyang Liu Please indicate by return email the full and correct funding information for your study and confirm the order in which funding contributions should appear. Please be sure to indicate whether the funders played any role in the study design, data collection and analysis, decision to publish, or preparation of the manuscript. **7)** Please send a completed 'Competing Interests' statement, including any COIs declared by your co-authors. If you have no competing interests to declare, please state "The authors have declared that no competing interests exist". Otherwise please declare all competing interests beginning with the statement "I have read the journal's policy and the authors of this manuscript have the following competing interests" **Reviewers' comments:**

Reviewer's Responses to Questions

**Comments to the Authors:**

Reviewer #1: In the manuscript "Analytical and computational solution for the estimation of SNP-heritability in biobank- scale and distributed datasets " by Qi et al., the authors describe an improved implementation of the randomized Haseman-Elston regression (RHE-reg) to estimate SNP heritability on biobank-scale data (i.e., genomic analysis with hundreds of thousands of samples genotyped with hundreds of thousands SNP). The methodology is extended to distributed data (i.e., analysis on genomic data coming from different sources under privacy). The work presents an analytical procedure to control the number of iterations of the RHE-reg, which has been shown to be a limitation of the RHE-reg. A free software is mentioned (https://github.com/gc5k/gear2), albeit without further information on the software. The work is based on previous works on the RHE-reg trying to address limitation on the number of iterations needed to provide with accurate solutions. Moreover, given the plethora of available software that can handle biobank-scale genomic data, the novelty of the work is questionable.

Overall, the topic is of interest to the readership of PLOS Computational Biology in the field of genetics/genomics. The manuscript is well written. Despite this, there are some critical points that need to be addressed:

The authors stated that the aim of the work is to estimate SNP-heritability for biobank-scale data. However, in the simulation the umber of samples was set to 1,000, 5,000, and 10,000, and the number of SNPs to 10,000, 50,000, and 100,000. None of this combinations is close to what is known as biobank-scale. Moreover, more details should be given for the simulation scenario, e.g., software used and relationship among samples (related or unrelated individuals were simulated?). Further, authors simulated that all SNPs were considered causal after a typical polygenic model. This is not enough information. Exact distributions used to sample SNP effects should be provided. Were the 10k SNPs included in the 50k and 100k sets? If yes, did they have the same effects? Did all SNP had an effect even small or the effect of some SNPs was set to zero? Values of heritability were set to 0, 0.1, and 0.25. Is there any reason that medium to high heritability was not tested? I would strongly recommend to test model performance also with h2 of 0.5, 0.75 and 1 (or close to 1). Regarding the case of distributed data, how the model performs in the case of unbalanced data across institutes?

Some extra information that is missing and is of interest is the capacity of the computer used to run the analysis, the time required to run each analysis and a comparison with at least two independent and well-known software, as a base-line, that can analyse biobank-scale genomic data. Moreover, as mentioned above, although a github repository is mentioned for the software, no information is provided on the software. It is not clear finally how many iterations are needed to run biobank-scale analysis. In the real data analysis, could you explain the reason to use only unrelated individuals?

The Discussion is too short. All Tables and Figures need to be self-explanatory. Regarding the analysis with the UK-Biobank data (ExData2) there are some discrepancies that need to be discussed. Overall, the standard errors of the h2 estimates are higher compared to those reported in Xu et al. For the trait “Age of primiparous women at birth of child” no estimates were provided. In some cases, there are considerable differences in h2 estimates between Xu et al and the current study, e.g., 0.42 vs 0.17 for the trait “Trunk predicted mass” and 0.73 vs 0.50 for the trait “Trunk predicted mass”. Could you provide with explanations?

Overall, I support the publication of this manuscript, but only after addressing all my comments and/or suggestions point by point. Thus, my conclusion is major revision.

Minor comments

• In all equations double check the correct use of “~” and “^” for the predicted and estimated values.

• L 44 “much faster than REML” – be more precise

• L 46 “recently a randomized” – I think that a work back in 2018 cannot be considered a s a recent work.

• L 63 “horizontal federated learning” – could you elaborate more?

• L 68 “a large B” – be more precise

• L 69 “boundaries of key statistics” – could you explain more?

• L 95 “of the square of each element” – replace with “of the square of each diagonal element

• L95 “We proved that” – citation is missing

• L 97 “correlations” – you mean pearson correlations?

• L 100 tr(Kc) could you provide with the space of the values for c?

• Equation 3 – denote “L”

• L 101 “of of L2,B” remove second “of”

• L 108 “random mating or little inbreeding“ – up to which degree of inbreeding? Be more precise

• L 114, what is a and b?

• L 115, explain μ in the equation

• L 118 replace “Tylor” with “Taylor”

• L 126 “sufficient iterations” – how many?

• L 127 “large enough” – do you mean going to infinity?

• L 152 in the equation explain i and j.

• L 202 “c institutes” – consider to change the letter “c” in order not to be confused with “c” used in previous equations

• L 204-205 “yν” and “Xν”, explain subscript “ν”

• L 206 explain “F”

• L 217 “Furthermore, if we have c covariates, and the covariate matrix W is of n x c dimensions” – double check this sentence

• L 258 “because = 1,000 was too small a sample size here” – what does this mean? Were there any convergence issues?

• L 265 “much higher h2 took a much greater B” – please be more precise

• Figure 2 -consider to use same y-axis scale for fair comparisons

• Figure 3 – explain the blue and red lines. Consider to change h2(B20/B50) to h2(B20) – h2(B50) or h2(B20) / h2(B50) etc. Why negative h2 values are reported?

• Figure 4 – what are the negative h2 values?

• L 337 “a high degree” – consider to change to “a relative high degree”, since the pearson correlation reported is 0f 0.77

• Figure 5 – what is the meaning of h2 estimates > 1?

• L 348 and 349 should 30 be replaced with 50?

• Figure 6 - what is the meaning of h2 estimates > 1?

• L 385 – 391 are coming “out of the blue” in the discussion. Consider to make a subheading.

Reviewer #2: In this manuscript, the authors developed an analytical solution for scalable estimator of SNP heritability. They conduct simulations and real data application to illustrate the accuracy of their methods. The manuscript is well structured. I have a few questions and suggestions for the authors, which I listed below.

Major:

1. Equations and Notation. I have several questions regarding the notation used throughout the manuscript. Please review the notation carefully to ensure consistency and clarity across the entire text.

a. It would be helpful to define all notations at their first occurrence. For instance, the symbol c in Equation (3), the distribution of z_b, and the variables listed in Table 1 (e.g., q, v) are not clearly defined.

b. Line 100: I believe the term m^c in the denominator of L_{c,b} should be removed, given that K is defined as XX^t / m. This also differs from Equation (10) in Wu and Sankararaman (2018). Please check for similar inconsistencies in Table 2 and elsewhere in the manuscript.

c. Line 158: I find the notation in Equation (11) confusing—particularly the use of tilde m_e, which appears to correspond to vertical RHE-reg in Table 2. Also, is the second part of Equation (11) meant to represent the variance of the estimator \hat{m_e}? If so, the corresponding entry in Table 2 should be updated accordingly. Please use hats (^) to indicate estimated values throughout the text.

d. Line 242: Should the expression be B/2 rather than 1/B?

2. It would strengthen the manuscript to include a comparison between your method and existing approaches, such as that of Wu (2018), in your simulations. Comparing metrics like mean squared error (MSE), computation time, and memory usage would be especially informative.

3. Figures. The figure captions should be more self-explanatory. Please clarify what the lines and data points represent.

a. Figure 2. I was expecting to see a direct comparison of MSE across the simulation scenarios (or with methods from prior work such as Wu's), which would be more informative than only comparing Lambda_1, Lambda_2, and Lambda_3.

b. Line 261-262, The caption states that Figure 2 shows how quickly B can reduce Lambda_2/B, but the y-axis shows only Lambda_2. Please clarify.

c. Line 262-266. Please elaborate on how Lambda_1, Lambda_2, and Lambda_3 relate to MSE, and what insights Figure 2 provides. The differing axis scales make it hard to interpret your conclusions.

Minor:

Line 287. “The fitted regression” — of what? It would be better to plot the data point and provide a more detailed explanation in the caption. Plus, the color legend of B is missing.

Figure 4. Including a 45-degree reference line would be helpful. Also, please explain what the data points represent.

Line 359. Please define what “split 1/2” and “split 2/1” refer to.

Reviewer #3: Qi at al developed an analytical and computational solution for estimating SNP-heritability at biobank-scale scale data, which is an extremely important problem but often challenged because of computational burden. I very much appreciate the authors’ theoretical effort to attack this important problem and the work is also impressive. My major comments are trying to help the presentation of the manuscript.

I found the current manuscript is not easy to understand. In the main text, there are a lot of mathematical formulas and derivations, but many steps have been missed, which are difficult to follow. I suggest the authors present the final formula with clear notation definitions and leaving detailed mathematical derivations in Supplementary Note. This should improve readability. I will try to give my specific suggestions in my comments. I also suggest the authors carefully check all the mathematical formulas and make sure they are correct.

Line 96-97, it will be good to point out where the prove of E(tr(K2)), and E(h ^2) are. It is also confused why E(h ^2) is still involved y? It will be good to add more details for the derivations, such as a Supplementary note.

Line 100, In equation (3), it will be better to introduce L_(c,B) first before writing the equation. From my calculation, the current definition of L_(c,B) leads to (〖E(L〗_(c,B))=1/m^c tr(K^c), which is inconsistent with the second equation. I think m^c only presents when K is writing as XX(T). I think the current definition of L_(c,B) (the first equation in equations (3)) does not have the term 1/m^c .

In addition, the equation var(L_(c,B) )=(2tr(K^2c))/B in equations (3) is not the same as the equation in Wu and Sankararaman (2018). I believe the authors’ equation is correct, but I think the authors should point out the inconsistence if this is true.

Again, line 110, L_(2,B) does not have 1/m2.

Equations (5), (6) and (7) need additional details. Again, a supplementary note will be good.

In equation (8), what is σ _(h^2 ) refers? There is no definition.

Line 129 and equation (10), why need z3? Should z3 be the same as z2? Confused.

Line 151, the title is “Estimation for the effective number of markers…”. But it is actually for estimating tr(K2).

Line 152, it should be E(tr(K2)) rather than tr(K2). The derivation should add additional detail, perhaps a Supplementary note.

Line 153, the fourth term summation should be m rather than n. I guess this term comes from ∑_(i≠j)^n▒∑_(k≠l)^m▒〖x_(i,k)^2 x_(j,l)^2 〗. Since i≠j, the expectation of this term should be n(n-1)m(m-1). I am not clear whether this difference will affect the conclusion.

Line 176, y _b is not defined.

Line 191, what is the numerator of Eq 5 refers?

Line 194, additional detail will be good.

Line 233-234, How LD was simulated? Was the LD randomly sample from 0, 0.2, …, 0.8?

Line 163-264, it states “Λ3 was seemed a less important… and vanished much faster than Λ2 “. Why it did vanish?

Figure 2 includes many symbols which cannot be recognized. It is possible due to the format.

Line 273, it states “the sample size Λ0=…”. Why does it refer to sample size?

Figure 3 only plotted the regression lines. Should be all the points be added so the readers can see how many data points were used in the regressions?

Line 336-338, it states the Pearson’s correlation coefficient 0.77. Should this correlation suggest the discrepancy between the current estimates and that by Xu at all?

I think the authors should add computational time for different B to see the improvement.

**Have the authors made all data and (if applicable) computational code underlying the findings in their manuscript fully available?**

Reviewer #1: Yes

Reviewer #2: Yes

Reviewer #3: Yes

PLOS authors have the option to publish the peer review history of their article (what does this mean? ). If published, this will include your full peer review and any attached files.

**Do you want your identity to be public for this peer review?** For information about this choice, including consent withdrawal, please see our Privacy Policy .

Reviewer #1: **Yes: ** Christos Dadousis

Reviewer #2: No

Reviewer #3: No

**Figure resubmission:**

**Reproducibility:**

---

## [Decision Letter · Decision Letter 1]

10 Sep 2025

PCOMPBIOL-D-25-00059R1

Analytical and computational solution for the estimation of SNP-heritability in biobank-scale and distributed datasets

PLOS Computational Biology

Dear Dr. Chen,

Thank you for submitting your manuscript to PLOS Computational Biology. After careful consideration, we feel that it has merit but does not fully meet PLOS Computational Biology's publication criteria as it currently stands. Therefore, we invite you to submit a revised version of the manuscript that addresses the points raised during the review process.

Please submit your revised manuscript within 30 days Nov 10 2025 11:59PM. If you will need more time than this to complete your revisions, please reply to this message or contact the journal office at ploscompbiol@plos.org. Please include the following items when submitting your revised manuscript:

We look forward to receiving your revised manuscript.

Kind regards,

Androniki Psifidi, DVM, PhD

Guest Editor

PLOS Computational Biology

Ilya Ioshikhes

Section Editor

PLOS Computational Biology

**Additional Editor Comments:**

Reviewer #1:

Reviewer #2:

Reviewer #3:

**Journal Requirements:**

1) We noticed that you used the phrase 'data not shown' in the manuscript. We do not allow these references, as the PLOS data access policy requires that all data be either published with the manuscript or made available in a publicly accessible database. Please amend the supplementary material to include the referenced data or remove the references.

2) Please amend your detailed Financial Disclosure statement. This is published with the article. It must therefore be completed in full sentences and contain the exact wording you wish to be published.

1) Please clarify all sources of financial support for your study. List the grants, grant numbers, and organizations that funded your study, including funding received from your institution. Please note that suppliers of material support, including research materials, should be recognized in the Acknowledgements section rather than in the Financial Disclosure

2) State the initials, alongside each funding source, of each author to receive each grant. For example: "This work was supported by the National Institutes of Health (####### to AM; ###### to CJ) and the National Science Foundation (###### to AM)."

3) State what role the funders took in the study. If the funders had no role in your study, please state: "The funders had no role in study design, data collection and analysis, decision to publish, or preparation of the manuscript."

4) If any authors received a salary from any of your funders, please state which authors and which funders..

3) Your current Financial Disclosure states, "Yes ↳ Please add funding details. national natural science foundation of China ↳ Please select the country of your main research funder (please select carefully as in some cases this is used in fee calculation). CHINA - CN".

However, your funding information on the submission form indicates different funders. 

Please indicate by return email the full and correct funding information for your study and confirm the order in which funding contributions should appear. Please be sure to indicate whether the funders played any role in the study design, data collection and analysis, decision to publish, or preparation of the manuscript.

**Reviewers' comments:**

Reviewer's Responses to Questions

**Comments to the Authors:**

Reviewer #1: Dear authors,

I would like to thank you very much for your work and apologise for any delays in the reviewing process.

There are few grammar errors that I believe will be corrected during final checks from PLOS computational biology.

At line 235 replace c institutes with s institutes.

Reviewer #2: Thank you for your efforts in revising the manuscript. My previous concerns have been greatly addressed. I just have one remaining question regarding new Figure 2.

Could you clarify why the scale of λ₂/B in the current Figure 2 reaches into the hundreds or thousands, while λ₂ was previously shown to be in the range of hundredths or lower (similar concern with λ3)? Also, the color coding appears off. In the fourth panel, the green dashed is supposed to indicate the mean along the x-axis, but there is no green dots on its right side. Panel 1 has the same issue.

As my previously comment, it would be better to draw the MSE value across the simulation scenarios.

And it would be better to add a color legend in Figure 3.

Reviewer #3: The authors addressed my concerns well. Thanks.

**Have the authors made all data and (if applicable) computational code underlying the findings in their manuscript fully available?**

Reviewer #1: Yes

Reviewer #2: Yes

Reviewer #3: Yes

PLOS authors have the option to publish the peer review history of their article (what does this mean? ). If published, this will include your full peer review and any attached files.

**Do you want your identity to be public for this peer review?** For information about this choice, including consent withdrawal, please see our Privacy Policy .

Reviewer #1: **Yes: ** Christos Dadousis

Reviewer #2: No

Reviewer #3: No

**Figure resubmission:**
---

## [Editor Report · Decision Letter 2]

29 Sep 2025

Dear Dr. Chen,

We are pleased to inform you that your manuscript 'Analytical and computational solution for the estimation of SNP-heritability in biobank-scale and distributed datasets' has been provisionally accepted for publication in PLOS Computational Biology.

Best regards,

Androniki Psifidi, DVM, PhD

Guest Editor

PLOS Computational Biology

Ilya Ioshikhes

Section Editor

PLOS Computational Biology

---

## [Editor Report · Acceptance letter]

PCOMPBIOL-D-25-00059R2

Analytical and computational solution for the estimation of SNP-heritability in biobank-scale and distributed datasets

Dear Dr Chen,

I am pleased to inform you that your manuscript has been formally accepted for publication in PLOS Computational Biology. Your manuscript is now with our production department and you will be notified of the publication date in due course.

With kind regards,

Anita Estes
